# BROWSECOMP-PLUS: A MORE FAIR AND TRANSPARENT EVALUATION BENCHMARK OF DEEP SEARCH AGENTS

## ABSTRACT

Deep search agents, which integrate large language models (LLMs) with search tools, have shown success in improving the effectiveness of handling complex queries that require iterative search planning and reasoning over search results. Evaluations on current benchmarks like BrowseComp relies on black-box live web search APIs, have notable limitations in (1) *fairness*: dynamic and opaque web APIs hinder fair comparisons and reproducibility of deep search agent methods; (2) *transparency*: lack of control over the document corpus makes it difficult to isolate retriever contributions. To address these challenges, we introduce BROWSECOMP-PLUS, a benchmark derived from BrowseComp, employing a fixed, carefully curated corpus. Each query in BROWSECOMP-PLUS includes human-verified supporting documents and mined challenging negatives, enabling controlled experimentation. The benchmark is shown to be effective in distinguishing the performance of various deep search agents. For instance, the fully open-sourced method Search-R1, when paired with the BM25 retriever, achieves 3.86% accuracy, whereas the GPT-5 achieves 55.9%. Integrating the GPT-5 with the Qwen3-Embedding-8B retriever further enhances its accuracy to 70.1% with fewer search calls. This benchmark allows comprehensive evaluation and disentangled analysis of deep search agents and retrieval methods, fostering insights into retrieval effectiveness, citation accuracy, and context engineering in deep search agents. Code and data will be released.

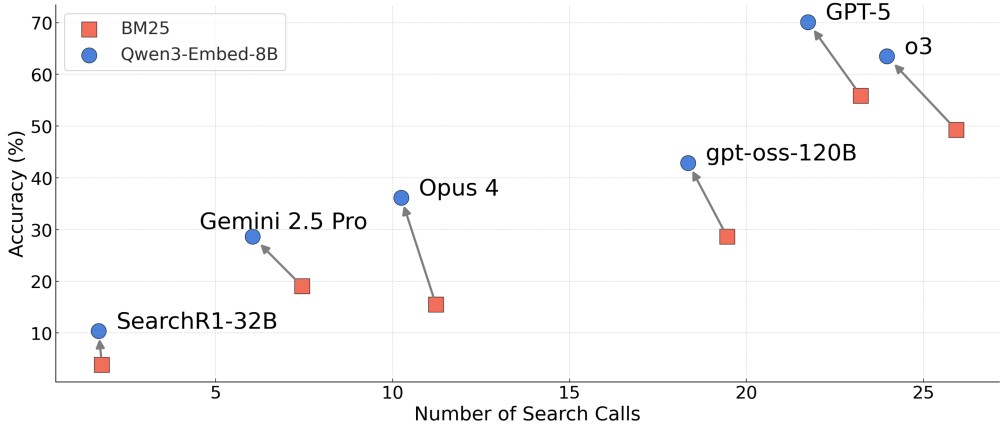

Figure 1: Accuracy vs. number of search calls for deep search agents with different retrievers. GPT-5, o3, gpt-oss are evaluated with high reasoning effort. The figure shows that **deep search agents mostly improve the final accuracy at a cost of more search calls,** whereas **better retrieval systems not only improve the overall accuracy but also reduce the number of search calls**. For reference, GPT-5 achieves 59.9% accuracy when evaluated using the Google Search API.

## 1 INTRODUCTION

Recent benchmarks for evaluating deep search agents, such as BrowseComp (Wei et al., 2025), have showcased the impressive capabilities of combining large language models (LLMs) with web search tools in solving complex, reasoning-intensive queries. These benchmarks typically provide sets of queries paired directly with answers, where agents are employed with live web search APIs to retrieve supporting documents in real time (Zhou et al., 2025; Chen et al., 2025). While this approach effectively assesses the end-to-end performance of deep search agents, it introduces several critical limitations that impede systematic analysis and evaluation of individual system components.

- **Fair Comparison of Deep Search Agents.** Current evaluations of deep search agents often conflate agent system performance with the effectiveness of their retrieval components, making it difficult to achieve fair and consistent comparisons across systems. This entanglement also severely undermines the reproducibility of experiments, which is a key requirement for rigorous evaluation (Voorhees, 2019).

- **Transparency of Retrieval Process.** The transparency of the retrieval process comes from two aspects: the retrieval algorithm and the target retrieval corpus. In the current evaluation pipelines, supporting documents are obtained through black-box web search APIs that operate over the entire internet, which are highly dynamic in content and consistently evolving over time. The lack of a controlled retrieval process hinders the evaluation of retrieval models' contribution to deep-research agents.

- **Accessibility**: The dependence on commercial web search APIs introduces substantial practical constraints, including high operational costs and variability in retrieval quality. These issues not only limit accessibility but also introduce unnecessary complexity and uncertainty.

To address these limitations and enable precise, reproducible, transparent, and component-focused evaluation of deep search agents, we introduce BROWSECOMP-PLUS, a new benchmark dataset. BROWSECOMP-PLUS extends the original BrowseComp dataset (Wei et al., 2025) by providing a fixed and curated corpus of documents specifically selected and verified by human annotators. Each query in BROWSECOMP-PLUS is accompanied by explicitly identified supportive documents and hard negative documents. This carefully collected document corpus allows researchers to evaluate the retrieval and LLM agent components independently, facilitating detailed analysis of each component's impact on the final answer quality. Additionally, by eliminating reliance on dynamic web APIs, BROWSECOMP-PLUS significantly reduces costs, enhances reproducibility, and improves the overall robustness of benchmarking in deep search agents.

To demonstrate the utility of BROWSECOMP-PLUS, we conduct comprehensive evaluations by pairing various open- and closed-source LLMs with a range of retrieval models on our curated corpus. This setup allows us to systematically analyze how different combinations affect answer quality and to identify where performance bottlenecks lie, whether in the retriever or the language model. We find that even when equipped with state-of-the-art retrievers, Deep-Research agents still face substantial challenges in consistently surfacing all necessary evidence, for reasoning-intensive queries. These findings motivate the need for evaluation frameworks that disentangle retrieval from reasoning, support fine-grained component analysis, and remain fully reproducible.

Furthermore, we extend our evaluation to test retrieval models directly on the original BrowseComp queries, an analysis that was previously infeasible due to the absence of a fixed corpus and grounded relevant document judgments. Our findings reveal that even state-of-the-art retrieval models struggle to retrieve relevant documents for these complex, reasoning-intensive queries.

In summary, our contributions are threefold:

- We present BROWSECOMP-PLUS, a fair and transparent benchmark for deep search agents, featuring a fixed, human-verified corpus with supporting and challenging negative documents.

- We provide the first systematic analysis of retrieval–agent interactions under controlled conditions, evaluating a broad range of retrievers and LLM-based agents.

- We release all benchmark data, evaluation scripts, and baselines to facilitate reproducible research and foster future advances in various dimensions to improve the deep-research system.

## 2 RELATED WORKS

### 2.1 DEEP SEARCH AGENT

Deep search agents conduct tasks through iterative query reasoning, search planning, and reflection on retrieved results (Asai et al., 2024), outperforming the traditional single-round retrieval-agumented generation paradigm (Lewis et al., 2020). Commercial closed-source models such as Gemini (Gemini 2.5 Team, 2025), Opus (Anthropic Team, 2024b), and o3 (OpenAI Team, 2025a), as well as open-source models like GPT-OSS (OpenAI Team, 2025b), allow access to external retrievers via tool-use APIs or MCP (Anthropic Team, 2024a). Recent research works such as Search R1 (Jin et al., 2025b) and WebSailor (Li et al., 2025), both based on the Qwen (Yang et al., 2025) model, leverage reinforcement learning to further enhance search tool capabilities. Fair evaluation of such agents, however, requires a fixed retriever system to make comparisons meaningful.

### 2.2 NEURAL RETRIEVAL

Neural retrieval methods, such as Dense Passage Retrieval (Karpukhin et al., 2020), encode queries and documents into dense vectors using transformer models, and perform retrieval through nearest-neighbor search (Douze et al., 2024). These methods have significantly improved retrieval effectiveness compared to traditional lexical-based methods like BM25 (Robertson, 1994). Recent improvements in neural retrievers include advanced training strategies such as continuous pretraining (Chen et al., 2024; Gao & Callan, 2022), data augmentation (Li et al., 2023; Ma et al., 2025b; Shao et al., 2025), integration of large language models as backbones (Ma et al., 2024; Wang et al., 2023), and LLM distillation techniques (Lee et al., 2024; Zhang et al., 2025). While retrievers are a critical component of deep search agents, the contribution of different retrievers to the overall performance of these agents remains underexplored.

### 2.3 DEEP SEARCH BENCHMARKS

Traditional benchmarks such as NaturalQuestions (Kwiatkowski et al., 2019) and TriviaQA (Joshi et al., 2017) have significantly contributed to evaluating retrieval and retrieval-augmented generation systems (Lewis et al., 2020; Karpukhin et al., 2020; Lin et al., 2024). However, these benchmarks primarily feature single-hop questions, which typically do not require multi-step reasoning or iterative retrieval. Although datasets like HotpotQA (Yang et al., 2018) offer multi-hop questions, the are shallow in depth ( 2 hops), and the corpus is limited to Wikipedia. To robustly evaluate deep search agents capable of complex reasoning and iteratively retrieve many turns, benchmarks requiring deep multi-turn query interactions are essential. BrowseComp (Wei et al., 2025) stands out as a benchmark explicitly designed for this purpose, offering complex queries paired with verifiable answers. Recent extensions of BrowseComp concepts, such as ZH-BrowseComp (Zhou et al., 2025) and MedBrowseComp (Chen et al., 2025), further expand to multilingual queries and domain-specific challenges. Mind2Web2 (Gou et al., 2025) on the other hand proposed to evaluating time-varied questions with agent-as-judge.

Existing benchmarks primarily focus on question-answer evaluations of integrated systems without standardized corpora, complicating comparative assessments of retrieval methodologies. BROWSECOMP-PLUS facilitates fair and comprehensive evaluations by providing human-verified corpus, expanding the classic Cranfield paradigm (Voorhees, 2002) to modern deep search agent evaluation.

## 3 BROWSECOMP-PLUS

### 3.1 PRELIMINARY: BROWSECOMP

The BrowseComp benchmark contains 1,266 challenging fact-seeking questions specifically designed to assess the capability of deep search agents to interactively and creatively navigate the web for complex, hard-to-find information (Wei et al., 2025). The questions are deliberately constructed to be difficult for both humans and LLMs, yet they feature verifiable, concise answers, enabling straightforward evaluation through simple answer matching. While effective and widely employed

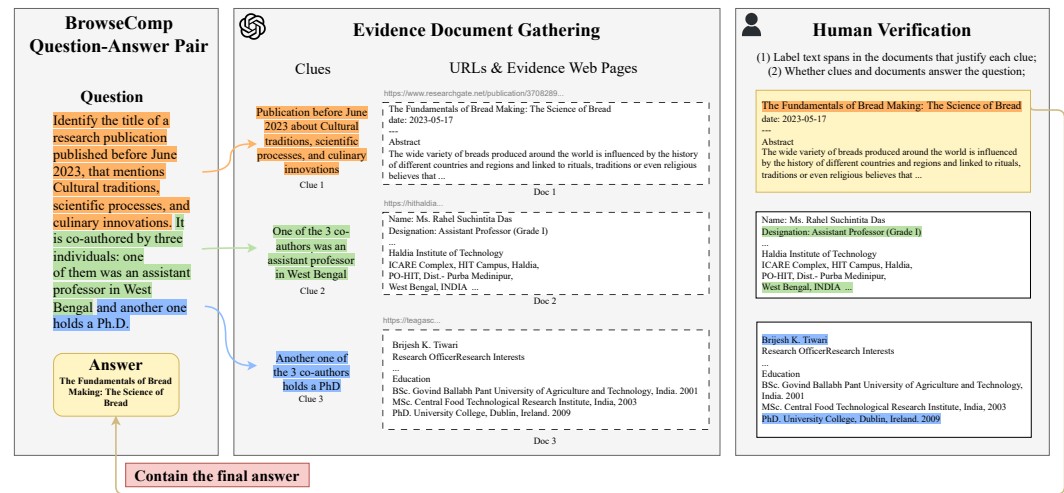

Figure 2: The two-stage pipeline of collecting evidence documents in the corpus (Section 3.2).

for end-to-end evaluation of deep search agents with web search access, this approach complicates the isolated measurement of retrieval effectiveness within these frameworks.

## 3.2 BUILDING THE DOCUMENT CORPUS

Constructing a corpus for BrowseComp questions is non-trivial. Three key challenges need to be addressed:

1. **Comprehensive coverage:** The corpus must provide complete evidence to support the entire reasoning chain required to answer each question.

2. **Retrieval difficulty:** The corpus should contain enough distracting negative documents so that search agents and retrievers are challenged in locating the correct evidence.

3. **Practical size:** The corpus should be large enough to yield reliable research insights, while avoiding overly-large computation costs for research purposes.

To meet these criteria, we curate evidence documents through a two-stage pipeline involving automated evidence mining followed by human verification, and perform hard-negative mining via web search to attach challenging, distracting documents to each query. The sections below describe this process in detail and present a 100k-document corpus that effectively supports the study of deep search agents.

### 3.2.1 EVIDENCE DOCUMENT GATHERING

The original BrowseComp dataset contains only question-answer pairs, without the URLs of the web pages that support these answers. To build a document collection with supporting evidence, the first step involves identifying relevant web pages for each question.

To achieve this, we leverage the OpenAI o3 model with web search enabled. Since the datasets intentionally make direct retrieval of relevant documents difficult, we adopt a *reverse-engineering* strategy: We provide the answer together with the question and instruct the model to search the web for pages that have evidence supporting the answers. We also ask the model to structure the output in a table format with three columns: (1) Clue: the part of the question to address; (2) URL: the web page link containing evidence supporting the clue; and (3) Evidence: the content from the web page that supports the clue. The purpose of this table format is to facilitate human annotators in verifying each clue and its corresponding web page in the next step. An example prompt for this step is provided in Appendix A.

Of the 1,266 original question-answer pairs in BrowseComp, the OpenAI o3 model fails to provide supporting evidence for 124 pairs, either due to output formatting errors or because the model abstains from answering due to low confidence. For the remaining 1,142 pairs, we scrape the URLs cited as evidence using Selenium,[1] and parse them with Trafilatura (Barbaresi, 2021). However, a combination of hallucinated URLs and scraping challenges prevents us from successfully scraping all of them. As a result, we exclude 137 question-answer pairs that contain at least one URL where we are unable to scrape, as missing a URL for a clue will make the question incomplete to answer.

This leaves us with 1,005 queries for the next stage: human verification.

### 3.2.2 Evidence Document Verification

In this stage, we aim to verify that the documents contain sufficient evidence for each clue in the questions. For each question-answer pair, we present human annotators with the output table from OpenAI o3 in the previous stage, with URLs replaced by the corresponding processed documents.

Annotators are asked to:

1. Confirm that each clue is sufficiently justified by the supporting documents. Instead of simply confirming the match, annotators must label the text spans in the documents that justify each clue, as this explicit step encourages high-quality verification.
2. Determine whether the combination of clues and supporting evidence enables a human to answer the *entirety* of the question correctly. For instance, if a query asks for an individual matching five characteristics, all five must be verifiable from the documents.

If the original output from OpenAI o3 fails to meet both criteria, annotators are instructed to revise the clues and search the web for additional supporting documents for at least 20 minutes, before concluding that the desired evidence documents cannot be collected.

In addition to constructing the evidence document set, annotators also label which documents directly contain the final answer; these are designated as *gold documents*. Note that a gold document is not defined merely by containing the ground-truth answer as an exact substring; in some cases, the answer is included in the document in an implicit way. For example, a question might ask for the number of publications by a particular author, with the ground-truth answer being "7". A gold document in this case could be the author's personal webpage listing their publications; while it may not contain the string "7" explicitly, it logically contains the answer. Similarly, there are many cases where the answer appears in the document in a variant form, such as a different date format or a paraphrased phrase, rather than an exact string match. Our goal in constructing the gold document set is to provide a more robust and semantically meaningful alternative to the simple substring-based approach in identifying documents that contain the final answer.

Figure 2 illustrates the complete evidence document collection process. A detailed example, including a screenshot of the labeling interface shown to human annotators, is provided in Appendix B.

For quality control, we sample each annotator's labeled data and cross-validate them among annotators, showing over 80% of agreement on average. Overall, of the 1,005 question-answer pairs from the previous stage, 830 passed human verification. The most common failure mode occurs when the documents provided by OpenAI o3 do not satisfy the two verification criteria, and human annotators are unable to gather sufficient additional evidence within a reasonable effort. In addition to these, we identify and exclude several other categories of problematic cases as detailed in Appendix C.

The entire labeling process involved 14 university student annotators and required over 400 hours of manual effort.

### 3.3 Hard Negative Mining

To ensure the collected corpus remains a reasonable size while still being challenging enough for search systems to identify correct answers among distracting documents, we mine hard negative documents via web search to form the corpus. This has proven to be effective in evaluating information retrieval systems using a sub-sampled corpus (Fröbe et al., 2025; Zhuang & Zuccon, 2022).

---

[1] https://www.selenium.dev/documentation

Specifically, we take each question from BrowseComp and prompt GPT-4o to break it down into simpler, self-contained sub-queries. On average, this results in about seven sub-queries per original query. Each sub-query is then sent to a Google Search API provider (SerpAPI), which returns up to 100 search results. We scrape these results using the same process used for collecting documents during positive example construction. We illustrate this hard negative document collecting process in Figure 4. The prompt used to create these sub-queries is provided in Appendix D.

## 3.4 FINAL CORPUS STATISTICS

After deduplicating the positive and negative documents collected as above, we arrive at a corpus of 100,195 documents, along with 830 queries. On average, each query contains 6.1 evidence documents, 76.28 negatives, and 2.9 gold documents. Each document averages 5179.2 words and 32296.2 characters.

## 4 EXPERIMENTS

### 4.1 EXPERIMENT SETUP

**Search Agents**    We list the agent baseline models in Appendix H.1. To perform agentic search with the LLMs, we provide the LLM with a retriever tool as tool use. We follow the original prompt from BrowseComp (Wei et al., 2025), which instructs the model to answer a given question along with a confidence estimate (expressed as a percentage). There are two revisions of the original prompts: (1) We explicitly prompt the LLM to use the provided tools to adapt to our custom search tool; (2) We instruct the model to cite the sources when generating the final answer, enabling the evaluation of citation quality. The complete prompt is included in Appendix E. We use this prompt across all models except Search-R1, which uses the prompt aligned with its original fine-tuning.

**Retriever**    We list the retriever baseline models in Appendix H.2. The retriever tool is set to retrieve the top $k = 5$ search results, where each result is truncated to the first 512 token of the corresponding document. This truncation is due to budget constraints, which prevent us from providing full document content. To assess the impact of this design choice, we analyze the distribution of the number of tokens required to include the ground-truth answer for each query. As illustrated in Figure 5 (b), when documents are truncated to the first 512 tokens, 86.5% of queries still contain the ground-truth answer in at least one of their gold documents. Further ablations exploring alternative tool configurations are discussed in Section 4.7.

### 4.2 EVALUATION METRICS

**Deep Search Agent Effectiveness**    We report end-to-end effectiveness of the deep search agents with three metrics: Accuracy, Recall, and Search Calls. Accuracy follows BrowseComp: an LLM-as-judge (GPT-4.1) compares the model's final answer against the ground truth using the evaluation prompt listed in Appendix F. Recall measures how many human-verified evidence documents the agent retrieved during its entire interaction. Search Calls is the average number of search API invocations per query. In addition, following BrowseComp, we compute calibration error using the confidence estimates produced by the search agents, in the same way as Humanity's Last Exam (Phan et al., 2025), measuring how closely a model's predicted confidence matches the actual accuracy of its predictions. For Search-R1, we do not report calibration error because the input and output format of this model are fixed without a confidence source output. Lastly, to understand whether the accuracy obtained by each agent stems from its agentic ability or merely its parametric knowledge, we also evaluate each LLM's accuracy when directly prompted with the question, without any retriever or external knowledge.

**Retrieval Effectiveness**    For evaluating retriever effectiveness, our BROWSECOMP-PLUS benchmark provides human-verified evidence documents and gold documents, along with a fixed test document collection, enabling evaluation under the Cranfield paradigm (Voorhees, 2019). Specifically, we follow standard TREC practice to create a query-document relevance label file[2] for both

---

[2]Known as a qrel file.

Table 1: End-to-end agent accuracy on BROWSECOMP-PLUS across LLMs and retrievers. All agents are prompted with the same tool-use prompt, except for Search-R1, which uses the prompt identical to its training.

| LLM | Retriever | Accuracy | Recall | Search Calls | Calibration Error |
|---|---|---|---|---|---|
| GPT-4.1 | None | 3.86% | *N/A* | *N/A* | 73.83% |
| | BM25 | 14.58% | 16.42% | 10.35 | 68.96% |
| | Qwen3-Embed-8B | 35.42% | 36.89% | 8.67 | 54.67% |
| o3 | None | 19.52% | *N/A* | *N/A* | 14.07% |
| | BM25 | 49.28% | 56.64% | 25.93 | 12.58% |
| | Qwen3-Embed-8B | 63.49% | 73.24% | 23.97 | 16.77% |
| GPT-5 | None | 26.18% | *N/A* | *N/A* | 24.57% |
| | BM25 | 55.90% | 61.70% | 23.23 | 13.50% |
| | Qwen3-Embed-8B | 70.12% | 78.98% | 21.74 | 9.11% |
| Sonnet 4 | None | 1.69% | *N/A* | *N/A* | 40.92% |
| | BM25 | 14.34% | 21.31% | 9.95 | 29.79% |
| | Qwen3-Embed-8B | 36.75% | 47.33% | 9.03 | 24.51% |
| Opus 4 | None | 2.42% | *N/A* | *N/A* | 11.95% |
| | BM25 | 15.54% | 22.96% | 11.22 | 22.00% |
| | Qwen3-Embed-8B | 36.14% | 50.84% | 10.24 | 12.79% |
| Gemini 2.5 Flash | None | 3.13% | *N/A* | *N/A* | 79.01% |
| | BM25 | 15.54% | 21.45% | 10.56 | 29.28% |
| | Qwen3-Embed-8B | 33.01% | 40.19% | 9.77 | 21.63% |
| Gemini 2.5 Pro | None | 7.47% | *N/A* | *N/A* | 76.72% |
| | BM25 | 19.04% | 22.81% | 7.44 | 51.58% |
| | Qwen3-Embed-8B | 28.67% | 35.31% | 6.04 | 44.08% |
| gpt-oss-120B-high | None | 3.13% | *N/A* | *N/A* | 48.89% |
| | BM25 | 28.67% | 35.50% | 19.45 | 46.48% |
| | Qwen3-Embed-8B | 42.89% | 52.63% | 18.35 | 40.34% |
| Qwen3-32B | None | 0.96% | *N/A* | *N/A* | 67.98% |
| | BM25 | 3.49% | 3.12% | 0.92 | 57.41% |
| | Qwen3-Embed-0.6B | 4.10% | 3.45% | 0.91 | 60.71% |
| | Qwen3-Embed-4B | 7.83% | 6.20% | 0.89 | 61.06% |
| | Qwen3-Embed-8B | 10.36% | 7.80% | 0.94 | 59.84% |
| | ReasonIR | 9.16% | 7.59% | 0.91 | 55.15% |
| SearchR1-32B | None | 0.48% | *N/A* | *N/A* | *N/A* |
| | BM25 | 3.86% | 2.61% | 1.78 | *N/A* |
| | Qwen3-Embed-0.6B | 5.66% | 5.30% | 1.73 | *N/A* |
| | Qwen3-Embed-4B | 9.40% | 7.90% | 1.68 | *N/A* |
| | Qwen3-Embed-8B | 10.36% | 10.17% | 1.69 | *N/A* |
| | ReasonIR | 9.43% | 8.37% | 1.74 | *N/A* |

evidence documents and gold documents separately, and then compute Recall@k and nDCG@k to assess the effectiveness of retrievers.

### 4.3 END-TO-END DEEP SEARCH AGENTS PERFORMANCE

Table 1 summarizes the overall deep search performance across different LLMs and retrievers. Proprietary models (GPT-4.1, o3, GPT-5, Sonnet-4, Opus-4, Gemini) demonstrate high answer accuracy, with OpenAI's GPT-5 achieving the highest accuracy (70.12%) when paired with the Qwen3-Embedding-8B retriever. Open-source models such as Qwen3-32B and SearchR1-32B lag behind proprietary models. With Qwen3-Embedding-8B as the retriever, Qwen3-32B achieves only 10.36% accuracy, compared to 35.42% for GPT-4.1 and 63.49% for o3. Notably, the only high-performing open-source model we studied is gpt-oss-120B in its high reasoning mode, which achieves 42.89% accuracy, surpassing Opus 4 when both are paired with Qwen3-Embedding-8B.

In general, closed-source agents call the search tool more frequently than open-source models. For instance, OpenAI's GPT-5 and o3 issue an average of more than 20 search calls per query, while Qwen3-32B and SearchR1-32B make fewer than 2, despite being explicitly prompted to use the tool. This reflects a test-time scaling effect: more exhaustive search correlates with better outcomes and aligns with prior findings that reasoning-intensive queries benefit from exploratory retrieval.

Table 2: Effectiveness of retrievers. The complete question is used as the query for all retrieval methods for fair comparison.

| Retriever | Recall@5 | Recall@100 | Recall@1000 | nDCG@10 |
|---|---|---|---|---|
| **Evidence Document Retrieval** | | | | |
| BM25 | 1.2 | 4.7 | 13.7 | 1.6 |
| jina-colbert-v2 | 5.7 | 18.1 | 35.7 | 7.9 |
| Qwen3-Embed-0.6B | 6.2 | 26.5 | 59.7 | 8.0 |
| Qwen3-Embed-4B | 9.8 | 40.2 | 71.8 | 14.0 |
| Qwen3-Embed-8B | 14.5 | 47.7 | 76.7 | 20.3 |
| ReasonIR-8B | 12.2 | 43.6 | 73.9 | 16.8 |
| **Gold Document Retrieval** | | | | |
| BM25 | 1.4 | 6.1 | 17.3 | 1.7 |
| jina-colbert-v2 | 6.6 | 20.4 | 39.7 | 6.8 |
| Qwen3-Embed-0.6B | 8.5 | 30.5 | 66.2 | 7.4 |
| Qwen3-Embed-4B | 13.0 | 47.3 | 77.0 | 13.6 |
| Qwen3-Embed-8B | 18.5 | 55.8 | 83.5 | 19.5 |
| ReasonIR-8B | 15.3 | 49.7 | 78.9 | 15.5 |

In the parametric-only setting where no retrieval of external knowledge is used, most LLMs show very limited accuracy. Only o3 and GPT-5 perform notably better, correctly answering about 20% of the questions; this may suggest that these models were trained on BrowseComp. When comparing across different LLM agents, this potential contamination is another important factor to keep in mind.

## 4.4 EFFECT OF RETRIEVAL QUALITY

A consistent trend observed across all models is that stronger retrieval leads to higher final accuracy. First, consider the retriever's effectiveness on our dataset. We evaluate retrieval performance using the original full queries, with results shown in Table 2. Compared to BM25, Qwen3-Embedding-8B and ReasonIR-8B achieve substantially higher recall and nDCG for both evidence document retrieval and gold document retrieval. Notably, we observe a model size scaling law within the Qwen3 embedding family; larger models consistently perform better, with Qwen3-Embedding-8B surpassing ReasonIR-8B at the 8B scale.

Now, as indicated in Table 1, replacing the BM25 retriever with a stronger retriever leads to significant accuracy gains across all LLM agents. For instance, OpenAI's GPT-5's accuracy improves from 55.9% to 70.12%, while Sonnet 4 and Opus 4 both achieve more than double their BM25 accuracy. This suggests a strong positive correlation between retrieval effectiveness and research agent accuracy.

Moreover, stronger retrievers potentially reduce the number of search calls. For most proprietary models, Qwen3-Embedding-8B reduces search calls by approximately 1–3 compared to BM25. This shows that better retrieval not only improves effectiveness (accuracy) but also efficiency (fewer tool calls). In Appendix K, we also report differences in proprietary agent API costs when using different retrievers. Agents using Qwen3-Embedding-8B incur lower costs due to fewer input and output tokens, further supporting the efficiency gains enabled by stronger retrieval.

In addition, Appendix I reports the coverage, average number, precision, and recall of the document citations attributed by the agent during answer generation, highlighting opportunities for future improvements in long-answer generation with proper evidence attribution via citations. We also assess the role of LLM rerankers as part of the retrieval module in Appendix J, showing the potential of further improving the effectiveness of deep search agents through reranking.

## 4.5 ORACLE RETRIEVAL

We evaluate effectiveness in an extreme oracle setting, where search agents are prompted with all labeled positive documents to answer the questions. In this setup, GPT-4.1 achieves an accuracy of 93.49%. This highlights two key points. First, it showcases the importance of the retriever: if the retriever is of perfect quality, search agents can attain substantially high accuracy on complex reasoning tasks in BROWSECOMP−PLUS, in contrast to the 14.58% baseline accuracy of GPT-4.1 when using BM25 as the retriever. Second, it validates the quality of the BROWSECOMP−PLUS

Table 3: OpenAI gpt-oss models in different reasoning effort settings

| LLM | Retriever | Accuracy | Recall | Search Calls | Calibration Error |
|-----|-----------|----------|--------|--------------|-------------------|
| gpt-oss-20B-low | BM25 | 4.11% | 5.36% | 1.89 | 40.89% |
| | Qwen3-Embed-8B | 13.37% | 17.37% | 1.87 | 36.34% |
| gpt-oss-20B-medium | BM25 | 16.39% | 21.96% | 13.72 | 41.78% |
| | Qwen3-Embed-8B | 29.88% | 41.31% | 13.64 | 35.99% |
| gpt-oss-20B-high | BM25 | 21.08% | 31.98% | 26.87 | 33.42% |
| | Qwen3-Embed-8B | 34.58% | 49.29% | 23.87 | 27.81% |
| gpt-oss-120B-low | BM25 | 9.52% | 8.54% | 2.06 | 43.59% |
| | Qwen3-Embed-8B | 24.94% | 22.50% | 2.21 | 40.96% |
| gpt-oss-120B-medium | BM25 | 23.73% | 27.02% | 9.73 | 45.78% |
| | Qwen3-Embed-8B | 37.59% | 43.45% | 9.64 | 41.77% |
| gpt-oss-120B-high | BM25 | 28.67% | 35.50% | 19.45 | 46.48% |
| | Qwen3-Embed-8B | 42.89% | 52.63% | 18.35 | 40.34% |

Table 4: Comparison of Qwen3-32B and GPT-4.1 with and without get-document tool, using Qwen3-Embedding-8B as retriever.

| Model | Accuracy | Search Calls | Get Document Calls | Calibration Error |
|-------|----------|--------------|--------------------|-----------------|
| GPT-4.1 | 35.42% | 8.67 | N/A | 54.67% |
| GPT-4.1 + get-doc | 43.61% | 10.03 | 1.85 | 54.28% |
| Qwen3-32B | 10.36% | 0.94 | N/A | 59.84% |
| Qwen3-32B + get-doc | 11.69% | 1.01 | 0.27 | 56.47% |

corpus itself: GPT-4.1, a non-reasoning model, is able to correctly answer 93.49% of questions using only the evidence documents in the corpus. For the remaining 6.51% of cases, human annotators reviewed each instance and confirmed that the answers are indeed answerable from the positive documents; the errors stem solely from GPT-4.1's failure to reason correctly.

## 4.6 IMPACT OF REASONING EFFORT

We evaluate how the reasoning effort of LLMs influences answer quality and retrieval behavior. To isolate this effect, we focus on the gpt-oss family, which offers three reasoning modes: *low*, *medium*, and *high*. As shown in Table 3, increasing the reasoning effort leads to substantial improvements in accuracy and recall across all model sizes and retrievers. For example, gpt-oss-20B with Qwen3-Embed-8B improves from 13.37% accuracy in *low* mode to 34.58% in *high* mode, along with a recall jump from 17.37% to 49.29%. Similarly, gpt-oss-120B with Qwen3-Embed-8B's accuracy rises from 24.94% to 42.89%. These gains, however, come with a trade-off: higher reasoning modes dramatically increase the average number of search calls (e.g., from ≈2 to ≈24 for gpt-oss-20B with Qwen3-Embed-8B), implying higher computational and latency costs. Interestingly, calibration error tends to decrease with higher reasoning effort, suggesting that the models become more aligned between confidence and correctness as they reason more extensively.

## 4.7 EFFECT OF DOCUMENT READING STRATEGY

In previous experiments, we always presented only the first 512 tokens of each retrieved document as a preview to the LLM during each round of search and reasoning, due to token budget constraints. However, in realistic deep search scenarios, agents often have access to a document reader tool that enables reading the full content of a document. To evaluate the potential benefit of such a tool, we conduct experiments with GPT-4.1 and Qwen3-32B, both with and without access to a whole-document reader (referred to as the get-document tool). Appendix G contains the revised prompt used when the get-document tool is added.

Results are shown in Table 4. For GPT-4.1, enabling the get-document tool improves accuracy from 35.42% to 43.61%, with a modest increase in search calls (from 8.67 to 10.03) and an average of 1.85 full-document reads per query, confirming that full-document access provides additional useful context that enhances decision-making. For Qwen3-32B, which performs worse overall, the benefit is

more modest. Accuracy improves slightly from 10.36% to 11.69%, and the number of get-document calls remains low (0.27 per query on average). This suggests that while the tool can help, the model's limited reasoning and tool-use ability constrain its ability to exploit the additional information.

## 5 CONCLUSION

We introduced BROWSECOMP-PLUS, a new benchmark designed to address the reproducibility, fairness, and transparency challenges in evaluating deep search agents. By grounding each query in a fixed, human-verified corpus containing both positive and hard-negative documents, our framework enables the independent and controlled assessment of retrieval and agent components.

Through extensive experiments, we demonstrate that retrieval quality substantially impacts both the effectiveness and efficiency of deep search agents. BROWSECOMP-PLUS provides a robust platform for probing these dynamics and paves the way for future research on co-optimizing retrievers and agents, improving out-of-distribution tool-use generalization, and advancing context engineering frameworks. By making our benchmark and baselines publicly available, we aim to catalyze the next generation of deep search agents.

## REPRODUCIBILITY STATEMENT

The primary motivation for constructing BROWSECOMP-PLUS is to enable fair and reproducible comparisons of deep search agents. To this end, we will release the constructed corpus, pre-built retrieval indexes, and one-click reproducible code for evaluating all combinations of deep search agents and retrievers presented in this work. In addition, we plan to open source the full execution traces of our experiments, since some baselines are expensive to reproduce (e.g., running Opus 4 on the 830 BROWSECOMP-PLUS queries can incur approximately USD $2,000). By releasing these traces, we hope to help lower barriers for future researchers and support more efficient development of deep search agents.

## ETHICS STATEMENT

The BROWSECOMP-PLUS dataset extends OpenAI's BrowseComp, which is released under the MIT license. The augmented corpus was obtained by scraping documents from publicly accessible web sources searched via a Google API provider. As the data is drawn solely from open web content, we assess the ethical and legal risks to be minimal.

## LIMITATION

BROWSECOMP-PLUS has several limitations that we acknowledge and hope future work can address. First, although the corpus is constructed through careful human verification, it is difficult to guarantee the absence of false negatives, relevant documents that are missing from the corpus. This limitation is inherent to all large-scale retrieval benchmarks, but it may still create a gap between our benchmark and ideal evaluation. Second, the initial evidence-gathering step uses an OpenAI model (o3) to propose candidate URLs, which may introduce bias toward distributions that are more easily surfaced by that model; although humans subsequently edited or replaced many documents, this potential bias should be noted. Third, BROWSECOMP-PLUS primarily evaluates textual evidence and does not fully capture the diversity of real-world web content, such as interactive pages, dynamic layouts, multimedia, or unparsed PDFs. Finally, in this work we focus on evaluation based on short, concluded answers and cited documents within long-form responses. Comprehensive evaluation of generated reports for complex, deep-search agent tasks remains an open direction for future work.

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

## A  OpenAI o3 Evidence Document Gathering Prompt

I will give you a question and a correct answer, and you are to search online for evidence that supports the answer. List the evidence you've used to justify this answer step-by-step, including their urls in your output. Your final list of urls should be in the order such that a human can visit them in order to justify the answer.

Question: {question}

Answer: {answer}

This is all the information you have to work with to produce the final list of urls. Format your answer in a table with 3 columns:
- clue: the clue mentioned in the question
- url: the http web url of the evidence you've found
- evidence: the content in the url page that supports the clue

## B  Labelling UI Example

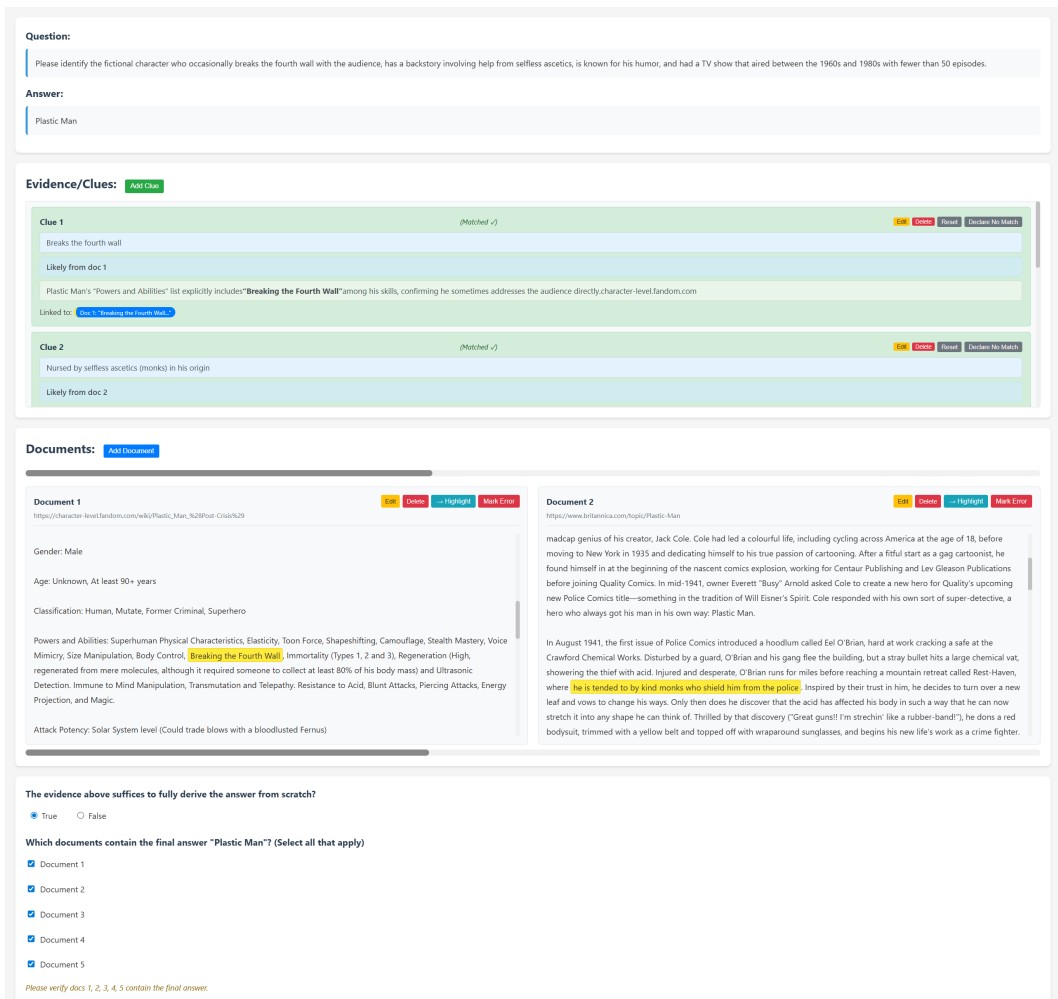

Figure 3: A screenshot of the annotation interface.

## C   PROBLEMATIC CASES

- BrowseComp Errors: During the verification process, we discover that some question-answer pairs in BrowseComp are inherently flawed. For example, one question asks for the name of a book whose author later returned to acting. Using the ground-truth answer, we can identify the intended book and its listed author. However, upon further investigation, we find that the individual who wrote the book and the one who returned to acting are two different people who happen to share the same name.

- Extensive Use of Google Maps: 42 queries in BrowseComp require distance-related information that explicitly prompt multiple calls to Google Maps. These are removed because high-quality documents discussing specific Google Maps distances between arbitrary locations are difficult to obtain. Moreover, scraping static snapshots of Google Maps pages to include in the corpus is not a valid substitute; answering such questions as intended should require agents to be augmented with access to the Google Maps API, rather retrieving from a corpus. However, this capability lies outside the scope of our objective to build a static, document-based dataset.

- Ambiguous or Non-Unique Answers: Some question-answer pairs are well-supported by documents, but suffer from ambiguity in the expected answer format or the existence of multiple valid answers. For instance, one question asks for the username of an individual who authored a specific story on an internet forum. While the ground-truth answer is correct, it is only one of three usernames credited as authors. We remove 13 such queries due to this kind of ambiguity.

## D   NEGATIVE MINING QUERY DECOMPOSITION PROMPT

You are an expert at breaking down complex, multi-part questions into simpler, self-contained subqueries.

Your task is to analyze the given question and decompose it into a series of smaller, more manageable subqueries that, when answered together, would provide all the information needed to answer the original question.

Guidelines:

1. Each subquery should focus on a single piece of information or concept
2. Subqueries MUST be completely self-contained and answerable independently - do not use pronouns or references like "this person", "the author", "these conditions", "they", "the movie", etc.
3. Each subquery should include all necessary context and constraints from the original query
4. Preserve all important details and constraints from the original query
5. Return only the subqueries as a JSON array of strings

Example:

Original: "Please identify the fictional character who occasionally breaks the fourth wall with the audience, has a backstory involving help from selfless ascetics, is known for his humor, and had a TV show that aired between the 1960s and 1980s with fewer than 50 episodes."

Subqueries: [ "Which fictional characters occasionally break the fourth wall with the audience?", "Which fictional characters have a backstory involving help from selfless ascetics?", "Which fictional characters are known for their humor?", "Which TV shows aired between the 1960s and 1980s?", "Which TV shows had fewer than 50 episodes? ]

Please decompose this query into subqueries:
{query}

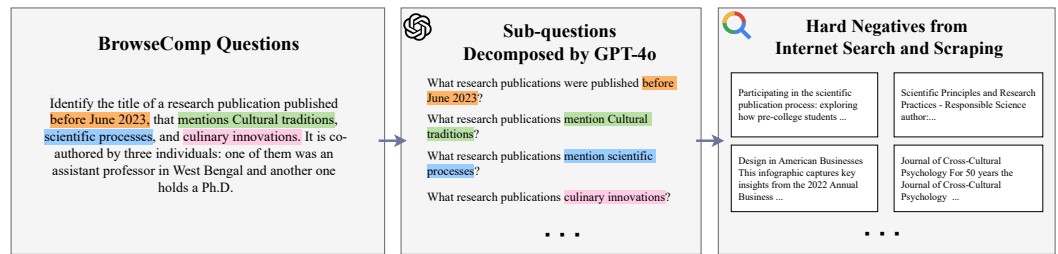

Figure 4: The pipeline of collecting hard negative documents in Section 3.3.

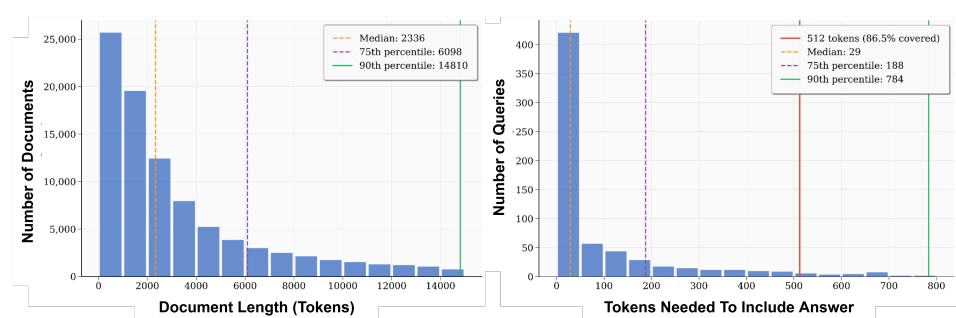

Figure 5: (a) Token distribution of corpus length, showing up to 90th percentile for display; (b) Distribution of tokens needed to include answer in gold documents per query, showing up to 90th percentile for display

# E  MAIN SEARCH PROMPT

You are a deep research agent. You need to answer the given question by interacting with a search engine, using the search tool provided. Please perform reasoning and use the tool step by step, in an interleaved manner. You may use the search tool multiple times.

Question: {Question}

Your response should be in the following format:
Explanation: {{your explanation for your final answer. For this explanation section only, you should cite your evidence documents inline by enclosing their docids in square brackets [] at the end of sentences. For example, [20].}}
Exact Answer: {{your succinct, final answer}}
Confidence: {{your confidence score between 0% and 100% for your answer}}

## F    EVALUATION PROMPT

Judge whether the following [response] to [question] is correct or not based on the precise and unambiguous [correct_answer] below.

[question]: {question}

[response]: {response}

Your judgement must be in the format and criteria specified below:

extracted_final_answer: The final exact answer extracted from the [response]. Put the extracted answer as 'None' if there is no exact, final answer to extract from the response.

[correct_answer]: {correct_answer}

reasoning: Explain why the extracted_final_answer is correct or incorrect based on [correct_answer], focusing only on if there are meaningful differences between [correct_answer] and the extracted_final_answer. Do not comment on any background to the problem, do not attempt to solve the problem, do not argue for any answer different than [correct_answer], focus only on whether the answers match.

correct: Answer 'yes' if extracted_final_answer matches the [correct_answer] given above, or is within a small margin of error for numerical problems. Answer 'no' otherwise, i.e. if there if there is any inconsistency, ambiguity, non-equivalency, or if the extracted answer is incorrect.

confidence: The extracted confidence score between 0|%| and 100|%| from [response]. Put 100 if there is no confidence score available.

## G    SEARCH PROMPT WITH GET-DOC

You are a deep research agent. You need to answer the given question by interacting with a search engine, using the search and get_document tools provided. Please perform reasoning and use the tools step by step, in an interleaved manner. You may use the search and get_document tools multiple times.

Question: {Question}

Your response should be in the following format:

Explanation: {{your explanation for your final answer. For this explanation section only, you should cite your evidence documents inline by enclosing their docids in square brackets [] at the end of sentences. For example, [20].}}
Exact Answer: {{your succinct, final answer}}
Confidence: {{your confidence score between 0% and 100% for your answer}}

## H    BASELINES

### H.1    LLM SEARCH AGENTS

We evaluate several representative commercial models with strong agentic search capabilities, ranging from the most advanced reasoning models to cost-effective ones: GPT-5, o3, GPT-4.1 (OpenAI Team, 2025a), claude-opus-4, claude-sonnet-4 (Anthropic Team, 2024b), gemini-2.5-pro, gemini-2.5-flash (Gemini 2.5 Team, 2025).

We also assess leading open-source efforts. This includes Qwen3-32B (Yang et al., 2025), a popular open-source reasoning LLM, and Search-R1 (Jin et al., 2025b;a), a model fine-tuned for agentic search based on the Qwen backbone. Specifically, we use the 32B checkpoint released in (Jin et al., 2025a). Finally, we evaluate the recent advanced gpt-oss-120B (OpenAI Team, 2025b), a reasoning LLM optimized for search tool usage that offers multiple reasoning effort settings, ranging from low to high.

## H.2 RETRIEVERS

In our study, we compared a range of retrieval methods from a traditional lexical baseline to modern state-of-the-art dense embedding retrievers:

- BM25 (Robertson et al., 1993): The classic sparse lexical retriever, which matches queries to documents based on term statistics.

- Qwen3-Embedding (Zhang et al., 2025): A dense embedding retriever, available in sizes 0.6B, 4B, and 8B, built on the Qwen3 foundation model family (Yang et al., 2025). It achieves state-of-the-art performance on retrieval benchmarks such as MTEB (Muennighoff et al., 2023).

- ReasonIR (Shao et al., 2025): A dense embedding specifically trained for reasoning-intensive retrieval via synthetic data generation, setting a new state-of-the-art on reasoning-intensive information retrieval benchmark BRIGHT (SU et al., 2025).

- Jina-ColBERT-v2 (Jha et al., 2024): A late-interaction retriever that trains ColBERTv2 (Santhanam et al., 2022) from a newer BERT backbone to support much longer contexts.

We use the Pyserini IR toolkit (Lin et al., 2021) to serve the BM25 retriever, the Tevatron dense retrieval toolkit (Ma et al., 2025a) to serve Qwen3-Embedding and ReasonIR, along with PyLate (Chaffin & Sourty, 2024) to serve Jina-ColBERT-v2.

## I CITATION QUALITY

Table 5 reports the coverage, average number, precision, and recall of the document citations attributed by the agent during answer generation. As the results show, although agents using BM25 issue more search calls, nearly all metrics are lower than those achieved with Qwen3-Embedding-8B. This indicates that documents returned by BM25 are less useful in the iterative deep research process, whereas Qwen3-Embedding-8B provides more relevant and informative documents.

## J EFFECT OF RERANKING

To evaluate the impact of reranking, we apply listwise reranking (Sun et al., 2023; Ma et al., 2023) over the top–20 and top–100 retrieved candidates using RankLLM (Sharifymoghaddam et al., 2025) with Qwen3-8B/32B and ReasonRank-7B/32B (Liu et al., 2025) models. The reranker operates with a sliding window of 20 candidates and a stride of 10, using a 16k-token context and a 16k-token thinking budget (output token count) to balance coverage and compute. Longer candidates are truncated to fit within the context window as needed.

Table 6 reports the effect of reranking after first-stage retrieval with Qwen3-Embed-8B, in the retrieval-only setting. For like-sized models, Qwen3 and ReasonRank perform similarly, with differences typically within 1 point. Overall, reranking yields sizable gains, improving Recall@5 by 8.4–24.0 points. With top-20 reranking, model size matters little (only ~2–3 points difference). Expanding the reranking candidate set to 100 improves all models, with larger gains for the 32B models, thereby widening the effectiveness gap between 8B and 32B models at higher rerank depths.

Table 7 reports the effect of integrating reranking into end-to-end performance of two search agents, GPT-4.1 and gpt-oss-20B (high reasoning effort), using Qwen3-Embed-8B as the first-stage retriever and Qwen3-8B to rerank the top 20 candidates. For both models, Accuracy (judged by GPT-4.1) and Recall improve substantially. This further indicates that reranking improves the precision and recall of retrieved evidence at higher ranks, helping the agent surface more relevant information.

Table 5: Per-query averages of citation coverage, citation count, precision, and recall for labeled evidence documents. Search-R1 is excluded because its fine-tuned outputs do not contain citations.

| LLM | Retriever | Coverage | Avg # Citations | Precision | Recall |
|---|---|---|---|---|---|
| GPT-4.1 | BM25 | 57.0% | 1.92 | 37.0% | 16.1% |
| | Qwen3-Embedding-8B | 79.2% | 2.54 | 58.5% | 28.2% |
| o3 | BM25 | 63.5% | 3.27 | 86.7% | 51.0% |
| | Qwen3-Embedding-8B | 78.0% | 3.51 | 91.8% | 56.2% |
| GPT-5 | BM25 | 94.9% | 3.89 | 71.8% | 51.3% |
| | Qwen3-Embedding-8B | 98.0% | 4.28 | 83.4% | 62.3% |
| Sonnet 4 | BM25 | 76.1% | 3.19 | 31.9% | 21.3% |
| | Qwen3-Embedding-8B | 90.7% | 4.19 | 52.4% | 39.9% |
| Opus 4 | BM25 | 74.9% | 3.03 | 35.1% | 22.3% |
| | Qwen3-Embedding-8B | 86.1% | 3.82 | 58.9% | 42.6% |
| Gemini 2.5 Flash | BM25 | 74.2% | 4.89 | 34.2% | 21.7% |
| | Qwen3-Embedding-8B | 89.2% | 4.75 | 51.5% | 35.1% |
| Gemini 2.5 Pro | BM25 | 53.9% | 3.03 | 52.1% | 31.4% |
| | Qwen3-Embedding-8B | 59.4% | 3.49 | 64.9% | 41.5% |
| gpt-oss-120B-high | BM25 | 62.5% | 3.55 | 50.8% | 31.5% |
| | Qwen3-Embedding-8B | 76.9% | 3.88 | 60.8% | 38.2% |
| Qwen3-32B | BM25 | 87.0% | 1.85 | 8.9% | 2.6% |
| | Qwen3-Embedding-0.6B | 90.1% | 1.79 | 8.7% | 2.5% |
| | Qwen3-Embedding-4B | 91.7% | 1.84 | 16.1% | 4.9% |
| | Qwen3-Embedding-8B | 90.2% | 1.78 | 20.0% | 6.6% |
| | ReasonIR | 95.8% | 1.74 | 18.0% | 5.7% |

Table 6: Effectiveness of rerankers with Qwen3-Embed-8B in retriever-only evaluation. The full question is used as the query in both stages. Reranking is applied to the top-20 and top-100 candidates. Scores in parentheses denote improvements over the base retriever ($\Delta$ vs. first stage).

| Reranker | Top–20 | | Top–100 | |
|---|---|---|---|---|
| | Recall@5 ($\Delta$) | nDCG@10 ($\Delta$) | Recall@5 ($\Delta$) | nDCG@10 ($\Delta$) |
| Qwen3-Embed-8B | 14.5 (−) | 20.3 (−) | 14.5 (−) | 20.3 (−) |
| **Evidence Document Retrieval** | | | | |
| ReasonRank-7B | 22.9 (+8.4) | 29.5 (+9.2) | 29.5 (+15.0) | 38.0 (+17.7) |
| Qwen3-8B | 23.3 (+8.8) | 30.0 (+9.7) | 29.6 (+15.1) | 37.7 (+17.4) |
| ReasonRank-32B | 24.9 (+10.4) | 32.1 (+11.8) | 34.4 (+19.9) | 43.8 (+23.5) |
| Qwen3-32B | 24.7 (+10.2) | 31.8 (+11.5) | 35.0 (+20.5) | 44.3 (+24.0) |
| **Gold Document Retrieval** | | | | |
| ReasonRank-7B | 28.7 (+10.2) | 28.9 (+9.4) | 36.8 (+18.3) | 37.1 (+17.6) |
| Qwen3-8B | 29.2 (+10.7) | 29.6 (+10.1) | 36.7 (+18.2) | 36.6 (+17.1) |
| ReasonRank-32B | 30.7 (+12.2) | 31.5 (+12.0) | 42.5 (+24.0) | 43.5 (+24.0) |
| Qwen3-32B | 30.5 (+12.0) | 31.3 (+11.8) | 42.2 (+23.7) | 43.0 (+23.5) |

## K  API Cost

Table 8 Shows the API costs of the experiments in Table 1.

## L  Future Work and Discussion

We believe that our BROWSECOMP−PLUS opens new avenues for advancing research in the Deep-Research area. BROWSECOMP−PLUS retains the challenging nature of the original BrowseComp while providing a more controlled and transparent experimental setup similar to early pivotal evalua-

Table 7: Effect of reranking on end-to-end agent performance. Qwen3-Embed-8B is used as the first-stage retriever and Qwen3-8B is used for reranking top 20 retrieved candidates.

| LLM | Retriever/Reranker | Accuracy | Recall | Search Calls | Calibration Error |
|---|---|---|---|---|---|
| GPT-4.1 | Qwen3-Embed-8B | 35.42% | 36.89% | 8.67 | 54.67% |
| | +Qwen3-8B | 47.11% | 51.46% | 8.77 | 49.86% |
| gpt-oss-20B-high | Qwen3-Embed-8B | 34.58% | 49.29% | 23.87 | 27.81% |
| | +Qwen3-8B | 40.24% | 57.98% | 21.98 | 21.47% |

Table 8: Overall API costs of proprietary agents for the experiments in Table 1.

| LLM | Retriever | Accuracy | Price (USD) |
|---|---|---|---|
| GPT-4.1 | BM25 | 14.58% | $106.96 |
| | Qwen3-Embed-8B | 35.42% | $89.81 |
| o3 | BM25 | 49.28% | $836.35 |
| | Qwen3-Embed-8B | 63.49% | $740.79 |
| GPT-5 | BM25 | 55.90% | $400.36 |
| | Qwen3-Embed-8B | 70.12% | $360.71 |
| Sonnet 4 | BM25 | 14.34% | $352.04 |
| | Qwen3-Embed-8B | 36.75% | $325.75 |
| Opus 4 | BM25 | 15.54% | $2,043.95 |
| | Qwen3-Embed-8B | 36.14% | $1,842.48 |
| Gemini 2.5 Flash | BM25 | 15.54% | $47.32 |
| | Qwen3-Embed-8B | 33.01% | $41.29 |
| Gemini 2.5 Pro | BM25 | 19.04% | $138.64 |
| | Qwen3-Embed-8B | 28.67% | $99.92 |

tion benchmarks like Natural Question (NQ) (Kwiatkowski et al., 2019) and HotpotQA (Yang et al., 2018). Like how NQ and HotpotQA have facilitated the design, comparison, and of modern neural QA systems, we hope that BROWSECOMP-PLUS will serve similar roles for Deep-Research agent studies. Here, we list some immediate research directions.

While our current work focuses on how different retrievers influence inference performance, a promising future direction is to examine the role of the retriever during agent optimization. For example, optimizing a search agent may be more challenging when paired with BM25 than with a modern embedding-based retriever, simply because BM25 surfaces fewer relevant documents. Understanding how retriever quality affects the learning dynamics of an agent remains an open question.

Another important extension is to study the agent's "out-of-distribution" tool-use capabilities. For instance, if an agent is optimized using a BM25 search tool, how well does its performance generalize when switched to an embedding-based search tool?

A more creative research could be an attempt on a breakdown of the commercial search engine. As much as a folktale, a commercial search solution employs tiered, composed, and multi-facet search solution. Is the LLM able to orchestrate a set of search tools to perform federated search (Wang et al., 2024), or even a sub-agent, to get quality results similar to those from Google?

A further direction is to design retrieval models that are tolerant of, or even adaptive to, a specific agent. In the Deep Research setting, the primary consumer of retrieved documents is no longer a human, but a tool-augmented LLM agent. This raises the possibility that retrieval models could be co-optimized with the agent for achieving overall answer accuracy, rather than developed and evaluated in isolation.

Finally, as shown in this work, an oracle retriever capable of surfacing gold or highly relevant documents can greatly improve accuracy. Such retrievers may also reduce the number of search iterations required, improving the overall efficiency of the research process. Developing high-

Table 9: Evidence document retrieval effectiveness on the Fineweb 10BT corpus.

| Retriever | Corpus | Recall@5 | Recall@100 | Recall@1000 | nDCG@10 |
|---|---|---|---|---|---|
| BM25 | Original | 1.2 | 4.7 | 13.6 | 1.6 |
| BM25 | Original + Fineweb | 2.2 | 8.0 | 19.4 | 3.1 |
| Qwen3-Embed-8B | Original | 14.5 | 47.7 | 76.7 | 20.3 |
| Qwen3-Embed-8B | Original + Fineweb | 11.6 | 37.6 | 64.2 | 16.4 |
| ReasonIR-8B | Original | 12.2 | 43.6 | 73.9 | 16.8 |
| ReasonIR-8B | Original + Fineweb | 8.6 | 30.7 | 56.3 | 11.8 |

Table 10: Accuracy of end-to-end search agents on our BROWSECOMP−PLUS original 100k corpus vs. FineWeb 10BT corpus.

| LLM | Retriever | Corpus | Accuracy |
|---|---|---|---|
| SearchR1-32B | BM25 | Original | 3.86% |
| | BM25 | Original + Fineweb | 4.72% |
| | Qwen3-Embed-8B | Original | 10.36% |
| | Qwen3-Embed-8B | Original + Fineweb | 8.33% |
| Qwen3-32B | BM25 | Original | 3.49% |
| | BM25 | Original + Fineweb | 5.42% |
| | Qwen3-Embed-8B | Original | 10.36% |
| | Qwen3-Embed-8B | Original + Fineweb | 7.11% |

precision retrieval systems for reasoning-intensive, complex queries could yield substantial benefits for real-world applications.

Overall, BROWSECOMP−PLUS serves as an ideal testbed for pursuing these directions, enabling systematic and fine-grained analyses of agent–retriever interactions within the Deep-Research paradigm.

## M EFFECT OF CORPUS SIZE

The corpus in BROWSECOMP−PLUS contains approximately 100K documents. While real-world agents often operate over much larger, web-scale corpora, we aim to assess whether our designed corpus size is sufficient to support valid experimental observations. To this end, we augment our benchmark corpus with the Fineweb-edu (Penedo et al., 2024) document collection (10 billion tokens),[3] deduplicated by URL. This expansion results in a significantly larger corpus of 9,771,311 documents—roughly 10 times larger than the original.

Table 9 shows retrieval performance before and after adding Fineweb documents. For BM25, retrieval effectiveness improves across all metrics, likely due to better inverse document frequency (IDF) estimation in the larger corpus, which strengthens BM25's lexical scoring.

In contrast, neural retrievers (Qwen3-Embedding-8B and ReasonIR-8B) show degraded performance on the Fineweb-augmented corpus. This drop is theoretically expected: the relative ranking of documents from the original small corpus remains unchanged, but the newly added Fineweb documents can now appear in the top ranks. Since these additional documents are unjudged, they are treated as non-relevant under standard TREC-style evaluation, inevitably lowering measured retrieval effectiveness.

It is important to note that lower retrieval scores for embedding models on Fineweb do not necessarily indicate worse final answers, some unjudged, top-ranked Fineweb documents may be "false negatives" that still provide useful evidence. However, as shown in Table 10, adding Fineweb does not improve answer accuracy for embedding-based retrievers. For example, Qwen3-32B with Qwen3-Embedding-8B drops from 10.36% to 7.11% accuracy.

---

[3]https://huggingface.co/datasets/HuggingFaceFW/fineweb-edu/viewer/sample-10BT

Overall, expanding the corpus size by a factor of 10 does not lead to different conclusions about the ranking or effectiveness level among the retrievers and LLM search agents, supporting our claim that the original 100K corpus offers both strong positive coverage and sufficient challenge for robust evaluation.

# N   USAGE OF LLM

ChatGPT is used during the writing to polish text (e.g., correct grammar) and format tables.

# O   SIGNIFICANT TEST OF MAIN RESULTS

In Table 6, we present the visualization of the significance test on the answer accuracy of each search agent integrated with different retrievers. The methods are ordered by their accuracy scores. The upper-right triangle indicates which pairwise comparisons reach the significance level of $p \leq 0.05$.

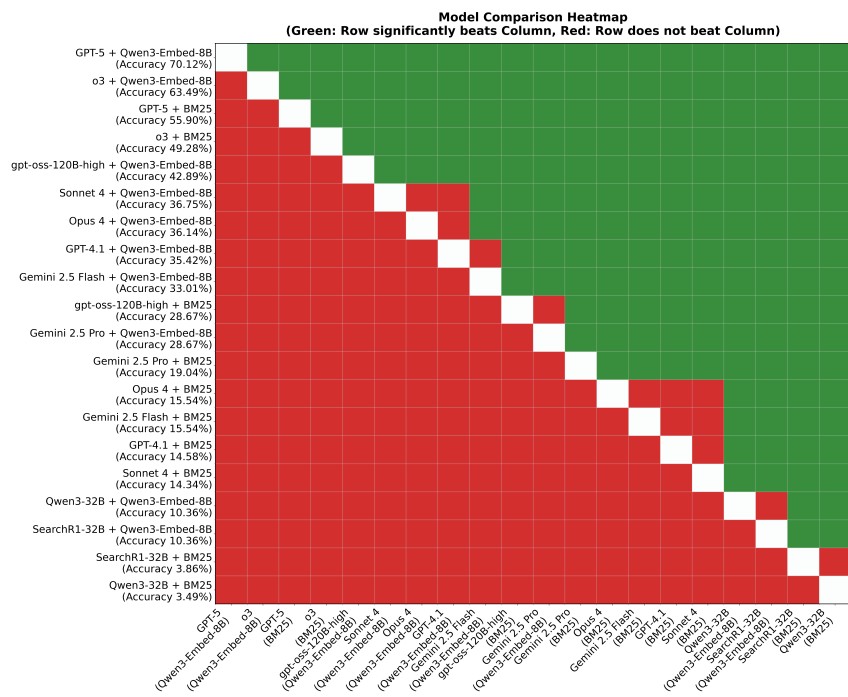

Figure 6: Pairwise McNemar's tests with Bonferroni correction were conducted at a significance level of ≤ 0.05. A green cell at Row (i), Column (j) indicates that the method in Row (i) performs significantly better than the method in Column (j).

# P   ANSWER ACCURACY WITH DIFFERENT JUDGMENT METHODS

In Table 11, we report answer-accuracy measurements using LLM-as-judge with GPT-4.1, Qwen3-32B, and substring matching. Across these evaluation methods, we observe consistent trends. Notably, the LLM-as-judge approach is more robust in handling cases where the predicted answers differ in format from the ground-truth labels.

| LLM | Retriever | Substring Match | GPT-4.1 Judge | Qwen3-32B Judge |
|---|---|---|---|---|
| GPT-4.1 | bm25 | 14.58 | 14.58 | 15.30 |
| GPT-4.1 | Qwen3-Embedding-8B | 34.46 | 35.42 | 36.39 |
| o3 | bm25 | 45.78 | 49.28 | 50.48 |
| o3 | Qwen3-Embedding-8B | 60.48 | 63.49 | 65.90 |
| Sonnet 4 | bm25 | 13.37 | 14.34 | 14.70 |
| Sonnet 4 | Qwen3-Embedding-8B | 33.73 | 36.75 | 37.35 |
| Opus 4 | bm25 | 15.18 | 15.54 | 15.54 |
| Opus 4 | Qwen3-Embedding-8B | 33.13 | 36.14 | 36.75 |
| Gemini 2.5 Flash | bm25 | 15.54 | 15.54 | 16.27 |
| Gemini 2.5 Flash | Qwen3-Embedding-8B | 31.45 | 33.01 | 34.58 |
| Gemini 2.5 Pro | bm25 | 17.71 | 19.04 | 19.88 |
| Gemini 2.5 Pro | Qwen3-Embedding-8B | 27.83 | 28.67 | 29.52 |
| Qwen3-32B | bm25 | 3.25 | 3.49 | 3.61 |
| Qwen3-32B | Qwen3-Embedding-0.6B | 4.22 | 4.10 | 4.22 |
| Qwen3-32B | Qwen3-Embedding-4B | 8.43 | 7.83 | 8.07 |
| Qwen3-32B | Qwen3-Embedding-8B | 9.76 | 10.36 | 10.72 |
| Qwen3-32B | ReasonIR | 8.67 | 9.16 | 9.28 |
| SearchR1-32B | bm25 | 3.86 | 3.86 | 4.11 |
| SearchR1-32B | Qwen3-Embedding-0.6B | 6.27 | 5.66 | 6.02 |
| SearchR1-32B | Qwen3-Embedding-4B | 10.60 | 9.40 | 9.28 |
| SearchR1-32B | Qwen3-Embedding-8B | 11.81 | 10.36 | 11.08 |
| SearchR1-32B | ReasonIR | 10.64 | 9.43 | 9.31 |
| oss-20b-low | bm25 | 3.51 | 4.11 | 3.99 |
| oss-20b-low | Qwen3-Embedding-8B | 11.93 | 13.37 | 14.10 |
| oss-20b-medium | bm25 | 15.54 | 16.39 | 16.87 |
| oss-20b-medium | Qwen3-Embedding-8B | 26.87 | 29.88 | 30.48 |
| oss-20b-high | bm25 | 19.76 | 21.08 | 21.45 |
| oss-20b-high | Qwen3-Embedding-8B | 31.93 | 34.58 | 35.06 |
| oss-120b-low | bm25 | 8.80 | 9.52 | 9.76 |
| oss-120b-low | Qwen3-Embedding-8B | 22.41 | 24.94 | 25.54 |
| oss-120b-medium | bm25 | 21.33 | 23.73 | 24.58 |
| oss-120b-medium | Qwen3-Embedding-8B | 33.49 | 37.59 | 38.55 |
| oss-120b-high | bm25 | 26.99 | 28.67 | 29.16 |
| oss-120b-high | Qwen3-Embedding-8B | 40.24 | 42.89 | 44.10 |
| GPT-5 | bm25 | 51.69 | 55.90 | 57.59 |
| GPT-5 | Qwen3-Embedding-8B | 65.18 | 70.12 | 71.69 |

Table 11: Comparison of accuracy measurement based on LLM-as-judge with GPT4.1, LLM-as-judge with Qwen3-32B, and sub-string Matching.

