# OpenReview forum: "BrowseComp-Plus: A More Fair and Transparent Evaluation  Benchmark of Deep-Research Agent"
_ICLR.cc/2026/Conference — ICLR 2026 Conference Withdrawn Submission_

### Official Review · Reviewer_aAkT · 2025-10-17

**Soundness:** 3
**Presentation:** 2
**Contribution:** 3
**Rating:** 4
**Confidence:** 4

**Summary:**

The paper introduced BROWSECOMP-PLUS, a deep research benchmark based on BROWSECOMP, with a provided retrieval corpus. Comparisons between multiple deep search systems and retrievers are also included.

**Strengths:**

- The paper proposed a fairer and more transparent deep research benchmark.
- Various LMs are included in the experiment.
- The authors are willing to share their full execution traces of their expensive experiments.

**Weaknesses:**

- The methodology and the dataset in the paper are not innovative enough. The queries in BROWSECOMP-PLUS is a subset of the queries in BROWSECOMP. It is also not clear if the methodology of building the document corpus can be applied to other scenarios.
- Missing details about methodology and evaluation. Please see Questions for more.

**Questions:**

I really appreciate the authors for the extensive evaluation of various LMs on deep search, particularly given that half of the models included in the experiments are closed-source. However, the paper would be strengthened by a more thorough justification of the **novelty** of the dataset itself and its construction methodology. The current discussion in the paper looks like simple application of existing methods to mine and append relevant and hard negative documents to the original BROWSECOMP dataset, supported by human annotation. Further clarification on the innovative aspects of this process is needed.

## Questions about Methodology:

* Section 3.2.1: Evidence documents were gathered by prompting an OpenAI model. This data collection method raises concerns about potential bias in the document corpus, favoring OpenAI models. The authors should address this potential bias.

* Section 3.2.2: Regarding human annotation, the paper states that annotators were “instructed to revise … and search … for at least 20 minutes.” What is the justification for this specific 20-minute time requirement?

* Section 3.2.2: The annotation of 1,005 queries, with each requiring at least 20 minutes, amounts to a minimum of 335 human hours for the verification step. The authors should provide details on the recruitment process for annotators and the quality assurance measures implemented to maintain high-quality annotations at this scale.

* Section 3.3: Hard negative documents were collected using the Google Search API for sub-queries. What steps were taken to ensure that the search results did not contain false negatives?

## Questions about Evaluation:

* Line 298: The paper states, “86.5% of queries still contain the ground-truth answer in at least one of their gold documents.” Was this figure calculated based on direct string matching, or did it also account for the “implication” cases mentioned in Lines 244-245?

* The authors have not specified how the LMs were integrated into the deep search pipeline. For instance, did the retrievers use the dataset's queries directly, or did they use search queries generated by the LMs? What factors determined the number of search calls?

* The oracle retrieval experiment in Section 4.5 is a valuable inclusion; however, I cannot find the results in the main paper or the appendix.

**Details Of Ethics Concerns:**

It is not clear if the crawled pages from Sections 3.2.1 and 3.3 have copyright issues or not. Also, the author did not discuss much about the human annotation process.

---

> ### Author Response · Authors · 2025-11-18
>
> We sincerely thank the reviewer for acknowledging BrowseComp-Plus’s utility in fairer agent evaluations, as well as the insightful comments and feedback. We provide point-by-point response as below.
>
> 1. **“Justification of the novelty of the dataset” (Q1)**
>
>     BrowseComp-Plus is, to the best of our knowledge, the first dataset to establish disentangled evaluation of search agents and retriever methods in deep search scenarios. Without a fixed and verified local corpus, two critical issues arise: (1) LLM agent researchers cannot fairly evaluate different search agent methods; (2) Retrieval researchers cannot perform standard indexing with proposed retrieval methods, preventing any development or evaluation of retrieval models and their interaction with search agents. With BrowseComp-Plus, we establish the motivation of improving retriever for search agents, and provide a reproducible testbed that paves the way for future researchers to improve both retrieval and search agents.
>
> 2. **“It is also not clear if the methodology of building the document corpus can be applied to other scenarios.” (W1)**
>
>     When constructing BrowseComp-Plus from BrowseComp, we did not use any tricks specific to the BrowseComp dataset. This same procedure can be applied to ground any other dataset into a corpus, given their query-answer pairs. For example, GAIA, another commonly used dataset to evaluate deep search agents, can follow the same corpus creation procedure for fairer evaluation across different methods. Our motivation for BrowseComp-Plus is not to simply rebrand BrowseComp, but to advocate for fair and transparent evaluation of deep search systems. We leverage BrowseComp to show that this data creation framework can deliver comprehensive evaluation over search agent and retriever’s effectiveness.
>
> 3. **“Evidence documents were gathered by prompting an OpenAI model. This data collection method raises concerns about potential bias in the document corpus, favoring OpenAI models” (Method Q1)**
>
>     Yes, using o3 to gather the initial evidence documents may introduce potential bias favoring OpenAI models. However, o3 only proposes the initial candidate set of URLs, which human annotators then select and edit. Starting from o3's initial set of 6,432 candidates, approximately 14.6% of the documents were modified (edited or replaced with new documents found by human annotators). Besides, the task o3 performs during data creation uses a reverse engineering strategy, "given question and answer, find evidence", which differs from the evaluation task of "given question, find evidence to answer." This distinction means o3 does not directly benefit when evaluated on the final benchmark.
>
>     From a benchmark perspective, BrowseComp-Plus aims to facilitate development of open search agent systems in academia. This potential bias does not affects the effectiveness of fair evaluation of open search agents or other proprietary agents.
>
>     However, we agree it is important to make this potential bias explicit, and we have added it to our limitations section. We greatly appreciate the reviewer's insight in raising this point, which presents a valuable improvement to our work.
>
> 4. **“What is the justification for the time limit of annotators searching for 20 minutes” (Method Q2)**
>
>     Given the large load of annotation, we cannot demand annotators to search indefinitely per question. Thus, we encourage the annotators to try to fix the issue with reasonable time and efforts rather than directly giving up.

---

> > ### Author Response · Authors · 2025-11-18
> >
> > 5. **“Details on the recruitment process for annotators and the quality assurance” (Method Q3)**
> >
> >     We recruited 14 university students in an information retrieval research group, 6 of which being current / completed PhD students specializing in information retrieval. Each annotator underwent approximately 1 hour of training on the labelling task on an internal development set prior to the real annotation on BrowseComp-Plus. Additionally, besides detailed text instructions, we created a 40-minute long demonstration video, covering many edge cases, which the annotators can constantly refer to. Further unsure cases were consolidated in a group channel, possibly relabelling prior cases for consistency. After the labelling process, 10 examples from each labeler were randomly sampled and discussed in the group channel, showing over 80% agreement. For the cases where a labeler made a mistake, the labeler was instructed to relabel all of their prior examples, avoiding similar mistakes. As noted in the paper, this process took well over 400 hours of manual effort.
> >
> >     The quality of the annotations can be shown through our parametric-only vs. oracle experiment. In both settings, we do not enable retrieval. In the parametric-only setting, we directly prompt GPT-4.1 with the question, and it achieves 3.86% accuracy (this is a new experiment shown in Table 1 of revised manuscript). In the oracle setting, we provide GPT-4.1 with all annotated evidence documents, and it correctly solves 93.49% of the questions. For the remaining 6.51% questions where it was incorrect, we have human verified that these were indeed answerable from the annotated evidence documents, and the error stems solely from GPT-4.1. Together, this demonstrates that our annotated documents suffice to ground the BrowseComp-Plus queries.
> >
> > 6. **“What steps were taken to ensure that the search results did not contain false negatives?” (Method Q4)**
> >
> >     There are indeed chances of false negatives that contain the same information as an evidence doc, but unlabelled. However, such false negatives is inevitable in all information retrieval corpora, because judging all documents in the corpora is infeasible, and is generally viewed as a tradeoff for attempting to mimic web-scale retrieval [1].
> >
> >     In our data creation process, our purpose is to ensure 1) the corpus contains a set of verified evidence document, so that the question is solvable by the corpus; 2) the corpus is challenging enough to differentiate advanced embedding models from traditional BM25.
> >
> >     To further verify that potential false negative do not impact the original intention of evaluation, we examined the negative documents and found that only 1.5% contain the ground truth answer.
> >
> >     [1] Corpus Subsampling: Estimating the Effectiveness of Neural Retrieval Models on Large Corpora.
> >
> > 7. **“86.5% of queries still contain the ground-truth answer in at least one of their gold documents. Was this figure calculated based on direct string matching?” (Evaluation Q1)**
> >
> >     The 86.5% still contains ground-truth answer refers to only substring matching, which does not include string variations (such as dates, names) and the “implication” cases mentioned in Lines 244-245. This means more than 86.5% of queries still contain the ground-truth answer.
> >
> > 8. **“how the LMs were integrated into the deep search pipeline. For instance, did the retrievers use the dataset's queries directly, or did they use search queries generated by the LMs? What factors determined the number of search calls?” (Evaluation Q2)**
> >
> >     The retrievers use search queries generated by the LLMs. We provide the retriever as a "search" function to the LLMs through tool use, and the LLMs autonomously decide when to call the tool or output an answer. This multi-turn tool use ability is built into the LLMs, following the API guidelines of providers like OpenAI and Claude. We did not write any logic or framework to intervene in the LLMs' decision-making. The number of search calls is the number of times the LLM uses the "search" tool before providing an answer. We will release our codebase with fully reproducible search agent scripts and the entire retriever setup.
> >
> > 9. **“The oracle retrieval experiment…I cannot find the results in the main paper or the appendix.” (Evaluation Q3)**
> >
> >     Our intent of having oracle experiment with human check is to verify that the corpus is able to fully answer all the questions, so we select GPT-4.1 to generate answer with oracle retrieval and examined the 54 queries (6.51% of all 830) where GPT-4.1 answered incorrectly, and we found that these questions were answerable from the provided documents, and it was incorrect due to GPT-4.1’s inability.
> >
> >
> > We sincerely appreciate the reviewer for the detailed feedback, and we hope that our response has addressed the main questions and concerns, while demonstrating the contributions of BrowseComp-Plus.

---

> ### Author Response · Authors · 2025-11-27
>
> Dear Reviewer aAkT,
> Thank you again for your thoughtful review and constructive suggestions. We have revised our paper accordingly and prepared a detailed rebuttal to address your concerns.
> As the discussion period is progressing, we would greatly appreciate it if you could let us know whether our response has adequately addressed your major concerns. We are happy to engage in further discussion and provide any additional clarification if needed.
> We look forward to hearing from you.

---

> > ### Comment · Reviewer_aAkT · 2025-11-27
> >
> > While the authors' comments have resolved most of my concerns, I remain hesitant regarding the novelty of this benchmark compared to previous RAG benchmarks. Specifically, the inclusion of a search corpus is not unique; similar approaches have been adopted in papers published earlier this year [1].
> >
> > A key benefit of including the retrieval corpus is the potential for deeper analysis, for example, quantifying how much of the generated output is grounded in the retrieved context. Given the transparency of your setting, I expected to see more extensive utilization of this data. Hence, I will maintain my current rating.
> >
> >
> > [1] Zora Zhiruo Wang, Akari Asai, Xinyan Velocity Yu, Frank F. Xu, Yiqing Xie, Graham Neubig, and Daniel Fried. 2025. CodeRAG-Bench: Can Retrieval Augment Code Generation?. In Findings of the Association for Computational Linguistics: NAACL 2025, pages 3199–3214, Albuquerque, New Mexico. Association for Computational Linguistics.

---

> > > ### Author Response · Authors · 2025-11-27
> > >
> > > Dear Reviewer aAkT,
> > >
> > > Thank you for the follow-up response and continued engagement with our work.
> > >
> > > **Regarding novelty**:
> > >
> > > To the best of our knowledge, BrowseComp-Plus is the first dataset to establish a disentangled evaluation of search agents and retrieval methods in deep search scenarios. Unlike traditional RAG settings, deep search requires the search agent to iteratively reason and interact with the retriever across many turns. The cited paper [1] operates in a traditional RAG setting, where the LLM aggregates search results in a single turn up front and focuses specifically on code generation. The scope and targeted task are fundamentally different from our work.
> > >
> > > We appreciate your suggestion to study **"how much of the generated output is grounded in the retrieved context"**, given the transparency our dataset offers.
> > >
> > > In fact, we have discussed this in the manuscript: In section 4.4, we studied retrieval effectiveness and their impact on end-to-end agents, and we also pointed to Appendix I, which comprehensively studies citation quality (coverage, citation count, citation precision and recall), quantitatively measuring how well and how much the generated output is grounded in the retrieved context. As you've mentioned, this is a meaningful insight that BrowseComp-Plus provides to current deep search agents.
> > >
> > > If you have additional suggestions for analyses we would be happy to provide further experiments or discussion.

---

### Official Review · Reviewer_4NMg · 2025-10-27

**Soundness:** 2
**Presentation:** 3
**Contribution:** 2
**Rating:** 2
**Confidence:** 4

**Summary:**

The paper introduces BrowseComp-Plus, a new benchmark designed to provide a more fair and transparent evaluation environment for search agents. BrowseComp-Plus employs a fixed, curated corpus and includes human-verified supporting documents as well as challenging negative samples. Through controlled experiments, the benchmark demonstrates the ability to differentiate performance across deep-research systems.

**Strengths:**

The paper addresses an important issue in evaluating search-based agents, namely the lack of fairness and transparency in existing benchmarks that rely on live web APIs. By introducing a fixed and curated corpus, the authors make a meaningful step toward reproducible and controlled evaluation. The inclusion of human-verified supporting documents and challenging negative samples strengthens the benchmark’s validity and difficulty, ensuring that evaluation results reflect true retrieval and reasoning capabilities.

**Weaknesses:**

1. The term Deep Research Agent in the title and whole paper seems an overclaim for the presented benchmark. At most, they can be referred to as Search Agents. Deep Research is a much more complex concept that involves tool use, evidence search, and synthesizing a comprehensive report at the end. That belongs to the category of “deep research,” which is also the focus of products like those from  OpenAI and Google Gemini. What this paper presents is more like search agents that only handle relatively simple QA-style problems. Therefore, it is an overclaim. Besides, since this work is based on BrowseComp by OpenAI, and the original paper only refers to Browsing Agents, the naming here should more appropriately be Search Agents, as this paper fixes the corpus database. Therefore, the current title and terminology are somewhat overstated and should be revised.

2. In the results section, it is suggested to add another type of baselines where each model is tested without using any external tools, relying only on the model’s own internal knowledge and skills. This would help observe whether there are other interesting phenomena and better understand the contribution of retrieval versus inherent model ability.

3. I understand that the authors evaluated Search-R1 using the open-source checkpoint to show that “the benchmark is effective in distinguishing the performance of deep research systems.” However, this is somewhat strange, because Search-R1 was not trained with such complex data. In the BrowseComp paper, the authors clearly stated that “the Deep Research model is trained on data that specifically teach the model to be good at BrowseComp tasks.” Therefore, for a benchmark paper, it would be better for the community if it could provide some training data and verify on several models that these training data are effective.

**Questions:**

See the weaknesses.

---

> ### Author Response · Authors · 2025-11-18
>
> We sincerely appreciate the reviewer for the insightful comments and suggestions. Please find below our point-by-point response:
>
> 1. **“The term Deep Research Agent in the title and whole paper seems an overclaim” (W1)**
>
>     Thank you for raising the question about "deep research" terminology. Please allow us to first clarify our initial purpose:
>
>     The core capability of a deep research agent is to iteratively search and reason to answer complex user questions. Our work focuses on evaluating this core ability, and several recent works use "deep research" to describe systems optimized for similar scenarios [1][2][3][4].
>
>     Additionally, in our evaluation, following BrowseComp’s setup, we prompt the LLM to generate a long answer before producing the final short answer. We evaluate citation quality within the long answer as a factor of overall quality.
>
>     However, we also agree with your concern regarding the "deep research" terminology. To be more rigorous, we have updated the title and corresponding discussion in the manuscript to "deep search agent" rather than "deep research agent" and emphasized that our main evaluation focuses on the search effectiveness capability of multi-turn interaction with retrievers.
>
>     [1] DeepResearcher: Scaling Deep Research via Reinforcement Learning in Real-world Environments. (EMNLP 2025)
>
>     [2] WebDancer: Towards Autonomous Information Seeking Agency. (NeurIPS 2025)
>
>     [3] Explore to Evolve: Scaling Evolved Aggregation Logic via Proactive Online Exploration for Deep Research Agents. (recent ArXiv)
>
>     [4] Open Data Synthesis For Deep Research. (recent ArXiv)
>
> 2. **“Search-R1 was not trained with such complex data” (W3)**
>
>     We agree that Search-R1 was not trained for such complex tasks. Search-R1 is a representative fully open-sourced research work that apply RL to improve the iterative search capability of LLM agent. We including the Search-R1 evaluation to show that the training in Search-R1 is not adequate to generalize to more complex task as in BrowseComp-Plus, calling for more fully open-sourced researches.
>
> 3. **“it would be better for the community if you could provide some training data” (W3)**
>
>     Please allow us to re-emphasize the importance of fair and transparent evaluation in deep search tasks, for both agents and retrievers.
>
>     For agents, prior works such as InForage [1] and WebShaper [2] have released training data and demonstrated that it is possible to effectively scale data creation for search agents. While this enables the community to build many models, we still cannot confidently assess the effectiveness of any training approach without *controlled retrieval*. As we show in BrowseComp-Plus, different retrievers can cause performance gaps of roughly 20 points across various agents.
>
>     For retrievers, BrowseComp-Plus is the first to explicitly define and evaluate the problem of retrieval for search agents. Before the community can solve this problem, it must be clearly articulated and measurable. BrowseComp-Plus highlights the importance of retrieval for search agents, and provides the first method for directly evaluating its effectiveness. Thus, while we do not provide training data, we provide the essential first step for future progress.
>
>     [1] Scent of Knowledge: Optimizing Search-Enhanced Reasoning with Information Foraging. (NeurIPS 2025)
>
>     [2] WebShaper: Agentically Data Synthesizing via Information-Seeking Formalization. (recent ArXiv)
>
> 4. **“It is suggested to add another type of baselines where each model is tested without using any external tools” (W2)**
>
>     We thank the reviewer for the great suggestion. This is indeed a valuable baseline to add and analyze. We have added a parametric-only setting without retrieval for all models in our main table, updating Table 1:
>
>     | LLM | Accuracy (%) | Calibration Error (%) |
>     | --- | --- | --- |
>     | GPT-4.1 | 3.86 | 73.83 |
>     | GPT-5 | 26.18 | 24.57 |
>     | o3 | 19.52 | 14.07 |
>     | Sonnet 4 | 1.69 | 40.92 |
>     | Opus 4 | 2.42 | 11.95 |
>     | Gemini 2.5 Pro | 7.47 | 76.72 |
>     | Gemini 2.5 Flash | 3.13 | 79.01 |
>     | gpt-oss-120B-high | 3.13 | 48.89 |
>     | Qwen3-32B | 0.96 | 67.98 |
>     | SearchR1-32B | 0.48 | N/A |
>
>     This ablation reveals that newer OpenAI models, o3 and GPT-5, achieve relatively high accuracy using parametric knowledge alone. This urges future agents to evaluate their accuracy both with and without search, giving the public a clearer sense of how much accuracy truly stems from agentic search ability. We sincerely appreciate this suggestion, as it strengthened our benchmark's fairness and transparency.
>
>
> We hope our response has covered the main concerns, and we want to thank the reviewer again for the suggested baseline, which greatly helped us in improving the paper. We always welcome further discussions during the reviewer-author discussion period.

---

> ### Author Response · Authors · 2025-11-27
>
> Dear Reviewer 4NMg,
> Thank you again for your thoughtful review and constructive suggestions. We have revised our paper accordingly and prepared a detailed rebuttal to address your concerns.
> As the discussion period is progressing, we would greatly appreciate it if you could let us know whether our response has adequately addressed your major concerns. We are happy to engage in further discussion and provide any additional clarification if needed.
> We look forward to hearing from you.

---

### Official Review · Reviewer_b7SQ · 2025-11-03

**Soundness:** 3
**Presentation:** 3
**Contribution:** 3
**Rating:** 4
**Confidence:** 3

**Summary:**

The authors propose BrowseComp-Plus, a benchmark for evaluating deep-research agents which can search and reason over web content.

Unlike BrowseComp that rely on live web APIs, BrowseComp-Plus uses a fixed, human-verified corpus with both supporting and hard-negative documents for the goal of having fair and reproducible benchmark. The dataset is built by combining model-guided evidence retrieval, human verification, and query-based negative mining.

Experiments test various agent–retriever pairs (like GPT-5, Qwen3-32B, BM25) on accuracy, recall, and efficiency. The authors aim was to show that BrowseComp-Plus offers a transparent, controlled benchmark for disentangling retrieval and reasoning in deep-research agents.

**Strengths:**

The paper is well-written and address an important problem related to Deep Research.

The paper proposes a large corpus to address multihop question answering with human-verified supporting facts and hard-distractor facts.

The authors evaluate multiple agent–retriever combinations across many experiment configurations including ablations related to open and closed source models.

**Weaknesses:**

It is not clear that this benchmark really measures deep research, as deep research question are more open-ended like "How can global climate policy reconcile economic growth with ecological preservation?". It feels more like a direct multi-hop question answering with closed form answers.

The idea and setup are very similar to Mind2Web2 (https://arxiv.org/abs/2506.21506), which also constructs a controlled environment for web reasoning and fact retrieval.

The paper does not explain how recall is handled when answers come from different websites that contain the same facts.

The use of LLM-as-a-judge may not be reliable. There should be some human evaluation to check if the model’s scores agree with human judgments (see this Deep Research Bench paper for inspiration https://arxiv.org/pdf/2506.11763).

There are no error bars or variance estimates, so we can’t tell if the differences between models are significant or just noise, and we can't tell if the llm-as-a-judge has large variance across runs.

It is not clear if the dataset covers diverse domains or if it is limited to a few common topics.

The finding that better retrieval gives higher accuracy is quite obvious as if the facts are not recalled it is impossible to achieve good solution. It is not clear why the llm is not able to answer the questions though if all the facts are recalled. This is something that needs to be studied.

The setup looks a lot like a retrieval-augmented generation (RAG) system. It’s unclear whether the agent really "browses" the web or just reads from a static collection of text files. Being able to navigate the web is a good challenge in itself. While the authors claim that web browsing is problematic due to changing content, the authors can always focus on timestamped posts that don't change over time and they can also use wayback machine (https://web.archive.org/).

True deep research should handle different file types (like PDFs, tables, or HTML pages), but this work seems limited to plain text.

The paper does not compare how well humans perform on these tasks, so we don’t know how difficult the benchmark really is. See GAIA for inspiration on how and why this is done (https://arxiv.org/abs/2311.12983).

There is no clear analysis of task difficulty (usually there are 3 levels like in GAIA) or model failure cases, which would help us understand where the models struggle.

**Questions:**

How do you distinguish deep research from multi-hop question answering in your benchmark, and what makes your tasks more complex than standard QA?

How is recall measured when multiple sources contain the same information are recalled if the website that the agent has acquired is not part of the groundtruth? and how do you deal with websites that have conflicting information, were all the items in the corpus checked for conflicts?

Did you validate the LLM-as-a-judge results with human evaluations to confirm that the scoring model is reliable?

Can you provide error bars or statistical tests to show whether the differences between models are significant?

How diverse are the domains and file types in your dataset, and could you expand the benchmark to include other content formats like PDFs, tables, or webpages?

---

> ### Author Response · Authors · 2025-11-18
>
> We sincerely thank the reviewer for the comprehensive feedback and your valuable time. Please find below our point-by-point response addressing each comment:
>
> 1. **“It is not clear that this benchmark really measures deep research, as deep research question are more open-ended…It feels more like a direct multi-hop question answering with closed form answers” (W1, Q1)**
>
>     Thank you for raising the question about "deep research" terminology. Please allow us to first clarify our initial purpose:
>
>     The core capability of a deep research agent is to iteratively search and reason to answer complex user questions. Our work focuses on evaluating this core ability, and several recent works use "deep research" to describe systems optimized for similar scenarios [1][2][3][4].
>
>     Additionally, in our evaluation, following BrowseComp’s setup, we prompt the LLM to generate a long answer before producing the final short answer. We evaluate citation quality within the long answer as a factor of overall quality.
>
>     However, we also agree with your concern regarding the "deep research" terminology. To be more rigorous, we have updated the title and corresponding discussion in the manuscript to "deep search agent" rather than "deep research agent" and emphasized that our main evaluation focuses on the search effectiveness capability of multi-turn interaction with retrievers.
>
>     [1] DeepResearcher: Scaling Deep Research via Reinforcement Learning in Real-world Environments. (EMNLP 2025)
>
>     [2] WebDancer: Towards Autonomous Information Seeking Agency. (NeurIPS 2025)
>
>     [3] Explore to Evolve: Scaling Evolved Aggregation Logic via Proactive Online Exploration for Deep Research Agents. (recent ArXiv)
>
>     [4] Open Data Synthesis For Deep Research. (recent ArXiv)
>
> 2. **“The idea and setup are very similar to Mind2Web2” (W2)**
>
>     Unlike BrowseComp-Plus, Mind2Web2 evaluates agents as entire systems, mixing retriever and agent contributions. BrowseComp-Plus enables explicitly quantified contributions of different retrievers, further supporting future development of retrieval methods for search agents. However, we appreciate the reviewer for mentioning Mind2Web2, and we will include it in related works.
>
> 3. **“The use of LLM-as-a-judge may not be reliable. There should be some human evaluation” (W4, Q3)**
>
>     Since the ground truth answers are factoid terms or short phrases, LLM-as-a-judge is commonly used and has been proven effective in previous widely used benchmarks such as Humanity's Last Exam, SimpleQA, and the original BrowseComp. BrowseComp-Plus uses the exact same evaluation setup matching BrowseComp and Humanity's Last Exam.
>
>     However, we agree that human validation is important. We verified all 830 responses for representative agents like GPT-5 and GPT-4.1. For GPT-5 + BM25, we found 5 judge errors and 4 debatable errors, totalling 9/830 ([**≈**](https://math.stackexchange.com/questions/864606/difference-between-%e2%89%88-%e2%89%83-and-%e2%89%85)1% error rate). For GPT-4.1, we observed 3 judge errors and 2 debatable errors, totalling 5/830 ([**≈**](https://math.stackexchange.com/questions/864606/difference-between-%e2%89%88-%e2%89%83-and-%e2%89%85)0.6% error rate).
>
>     We also evaluated using "substring match" (response contains answer) and LLM-as-judge from another LLM (Qwen3), as shown in the table below. These evaluation methods show strong consistency, and upon human comparison, LLM-as-judge performs better at handling cases where answers have format mismatches with ground truth. (We show the results of LLM with BM25 retriever below. Full results have been added Appendix-P in the revised manuscript).
>
>     | LLM | Substring Match (%) | GPT-4.1 Judge (%) | Qwen3-32B Judge (%) |
>     | --- | --- | --- | --- |
>     | GPT-4.1 | 14.58 | 14.58 | 15.3 |
>     | o3 | 45.78 | 49.28 | 50.48 |
>     | GPT-5 | 51.69 | 55.9 | 57.59 |
>     | Sonnet 4 | 13.37 | 14.34 | 14.7 |
>     | Opus 4 | 15.18 | 15.54 | 15.54 |
>     | Gemini 2.5 Flash | 15.54 | 15.54 | 16.27 |
>     | Gemini 2.5 Pro | 17.71 | 19.04 | 19.88 |
>     | oss-120b-high | 26.99 | 28.67 | 29.16 |
>     | Qwen3-32B | 3.25 | 3.49 | 3.61 |
>     | SearchR1-32B | 3.86 | 3.86 | 4.11 |

---

> > ### Author Response · Authors · 2025-11-18
> >
> > 4. **“How is recall measured when multiple sources contain the same information are recalled if the website that the agent has acquired is not part of the groundtruth? and how do you deal with websites that have conflicting information, were all the items in the corpus checked for conflicts?” (W3, Q2)**
> >
> >     There are indeed chances of false negatives that contain the same information as an evidence doc, but unlabelled. However, such false negatives is inevitable in all information retrieval corpora, because judging all documents in the corpora is infeasible, and is generally viewed as a tradeoff for attempting to mimic web-scale retrieval [1].
> >
> >     In our data creation process, our purpose is to ensure 1) the corpus contains a set of verified evidence document, so that the question is solvable by the corpus; 2) the corpus is challenging enough to differentiate advanced embedding models from traditional BM25.
> >
> >     To further verify that potential false negative do not impact the original intention of evaluation, we examined the negative documents and found that only 1.5% contain the ground truth answer.
> >
> >     Regarding potential conflicting information, this is indeed possible in our corpus. However, when searching over on the real web, there will also be such conflicting information. We believe it is a core ability of the search agent to reason and find the expected information among distractions.
> >
> >     [1] Corpus Subsampling: Estimating the Effectiveness of Neural Retrieval Models on Large Corpora.
> >
> > 5. **“Can you provide statistical tests to show whether the differences between models are significant?” (W5, Q4)**
> >
> >     Please see the table below for pairwise McNemar’s test at p ≤ 0.05 for the main table (Table 1), where the Outperform column in row i denotes all rows that row i outperform significantly.
> >
> >     |  | LLM | Retriever | Accuracy | Outperform |
> >     | --- | --- | --- | --- | --- |
> >     | (1) | GPT-5 | qwen3-8 | 70.12% | 2,3,4,5,6,7,8,9,10,11,12,13,14,15,16,17,18,19,20 |
> >     | (2) | o3 | qwen3-8 | 63.49% | 3,4,5,6,7,8,9,10,11,12,13,14,15,16,17,18,19,20 |
> >     | (3) | GPT-5 | bm25 | 55.90% | 4,5,6,7,8,9,10,11,12,13,14,15,16,17,18,19,20 |
> >     | (4) | o3 | bm25 | 49.28% | 5,6,7,8,9,10,11,12,13,14,15,16,17,18,19,20 |
> >     | (5) | gpt-oss-120B-high | qwen3-8 | 42.89% | 6,7,8,9,10,11,12,13,14,15,16,17,18,19,20 |
> >     | (6) | Sonnet 4 | qwen3-8 | 36.75% | 9,10,11,12,13,14,15,16,17,18,19,20 |
> >     | (7) | Opus 4 | qwen3-8 | 36.14% | 9,10,11,12,13,14,15,16,17,18,19,20 |
> >     | (8) | GPT 4.1 | qwen3-8 | 35.42% | 10,11,12,13,14,15,16,17,18,19,20 |
> >     | (9) | Gemini 2.5 Flash | qwen3-8 | 33.01% | 10,11,12,13,14,15,16,17,18,19,20 |
> >     | (10) | gpt-oss-120B-high | bm25 | 28.67% | 12,13,14,15,16,17,18,19,20 |
> >     | (11) | Gemini 2.5 Pro | qwen3-8 | 28.67% | 12,13,14,15,16,17,18,19,20 |
> >     | (12) | Gemini 2.5 Pro | bm25 | 19.04% | 13,14,15,16,17,18,19,20 |
> >     | (13) | Opus 4 | bm25 | 15.54% | 17,18,19,20 |
> >     | (14) | Gemini 2.5 Flash | bm25 | 15.54% | 17,18,19,20 |
> >     | (15) | GPT 4.1 | bm25 | 14.58% | 17,18,19,20 |
> >     | (16) | Sonnet 4 | bm25 | 14.34% | 17,18,19,20 |
> >     | (17) | Qwen3-32B | qwen3-8 | 10.36% | 19,20 |
> >     | (18) | SearchR1-32B | qwen3-8 | 10.36% | 19,20 |
> >     | (19) | SearchR1-32B | bm25 | 3.86% |  |
> >     | (20) | Qwen3-32B | bm25 | 3.49% |  |
> > 6. **“How diverse are the domains and file types in your dataset, and could you expand the benchmark to include other content formats like PDFs, tables, or webpages?” (W6, W9, Q5)**
> >     - Domain Diversity: The original BrowseComp classifies itself into 10 diverse domains. To validate that our corpus also covers these 10 diverse domains, we ran GPT-5-nano to classify each of the 100k docs, and obtained the following distribution:
> >
> >     | Domain | Percentage |
> >     | --- | --- |
> >     | Science and Technology | 17.20% |
> >     | History | 17.04% |
> >     | Other | 15.19% |
> >     | TV Shows and Movies | 12.29% |
> >     | Art | 8.66% |
> >     | Sports | 8.57% |
> >     | Music | 8.22% |
> >     | Politics | 6.82% |
> >     | Video Games | 3.40% |
> >     | Geography | 2.62% |
> >     - Other file formats: BrowseComp-Plus in fact already has many file formats: of the 5064 evidence documents needed to answer the 830 queries, 1475 of them contain tables, and 204 of them are parsed from PDFs. We did not intentionally include unparsed PDFs as we focus on evaluating the core reasoning + search ability of search agents.

---

> > > ### Author Response · Authors · 2025-11-18
> > >
> > > 7. **“The paper does not compare how well humans perform on these tasks, so we don’t know how difficult the benchmark really is” (W10)**
> > >
> > >     The original BrowseComp contains a difficulty analysis where humans gave up on 70% of the questions after spending ≥ 2 hours. GPT-5 achieves 54.9% accuracy on BrowseComp and 55.9% with BM25 on BrowseComp-Plus, reflecting the difficulty level. Since GPT-5 achieves this accuracy with over 20 rounds of search on average, it is impractical to have humans conduct that many search rounds. Additionally, scores from open-source methods like Search-R1 also demonstrate that our benchmark is challenging.
> > >
> > > 8. **“There is no clear analysis of model failure cases, which would help us understand where the models struggle…It is not clear why the llm is not able to answer the questions though if all the facts are recalled” (W7, W11)**
> > >
> > >     Following the reviewer’s suggestion, we conducted an error analysis for the failure cases on two representative proprietary and open-weight models, where we studied whether the runs retrieved at least 1 gold document throughout its search process: (1) If the run does not contain gold doc, it reflects the weakness in search-planing and retriever effectiveness; (2) otherwise, the run has encountered a gold doc, and the error reflects the agent’s weakness in evidence aggregation.
> > >
> > >     | LLM | Retriever | Accuracy | Success Contains Gold | Failure Contains Gold |
> > >     | --- | --- | --- | --- | --- |
> > >     | GPT-5 | BM25 | 51.69% | 99.57% | 38.52% |
> > >     | oss-120b-high | BM25 | 26.99% | 99.16% | 22.13% |
> > >     | GPT-5 | Qwen3-Embed-8B | 65.18% | 99.66% | 56.45% |
> > >     | oss-120b-high | Qwen3-Embed-8B | 40.24% | 98.88% | 32.70% |
> > >
> > >     As shown above, when using BM25, most of the failures could not retrieve gold documents, compared to the successes. When using Qwen3-Embed-8B, the failures with golds increased with accuracy, suggesting that a higher portion of the remaining failures are due to evidence aggregation.
> > >
> > >     This evidence aggregation failure also relates to the phenomenon where GPT-4.1 is unable to answer 6.51% of the questions correctly even when given all the evidence.
> > >
> > >     In addition, we are releasing the run files of all our experiments, aiming to further help the potential qualitative error analysis.
> > >
> > > 9. **“The setup looks a lot like a retrieval-augmented generation…better retrieval gives higher accuracy is quite obvious…can use wayback machine” (W7, W8)**
> > >
> > >     Recent advancements in LLMs for search raise an open question: "Will a weak retriever suffice as NLU and NLG models rapidly become stronger?" [1]. In the deep search scenario, a valid hypothesis is whether a strong search agent (e.g., GPT-5) with iterative search and reasoning can bridge the gap between a weaker but effective retriever (BM25) and a strong LLM retriever (Qwen3-8B). Our experiments **quantitatively** demonstrate this gap, showing that retrieval plays a crucial role and revealing new retrieval-related findings specific to multi-turn search agent tasks.
> > >
> > >     Importantly, we want to emphasize that BrowseComp-Plus is, to the best of our knowledge, the first dataset to establish disentangled evaluation of search agents and retriever methods in deep search scenarios. Without a fixed and verified local corpus, retrieval researchers cannot perform standard indexing with proposed retrieval method, which prevents any development or evaluation of retrieval models and their interaction with search agents. This makes the Wayback Machine infeasible for our purpose.
> > >
> > >     [1] Precise Zero-Shot Dense Retrieval without Relevance Labels
> > >
> > >
> > > We hope our response has addressed your main questions and concerns. We greatly appreciate your suggestions in helping us improve the paper, and we welcome further discussions during the reviewer-author discussion period.

---

> ### Author Response · Authors · 2025-11-27
>
> Dear Reviewer b7SQ,
> Thank you again for your thoughtful review and constructive suggestions. We have revised our paper accordingly and prepared a detailed rebuttal to address your concerns.
> As the discussion period is progressing, we would greatly appreciate it if you could let us know whether our response has adequately addressed your major concerns. We are happy to engage in further discussion and provide any additional clarification if needed.
> We look forward to hearing from you.

---

### Note · Authors · 2026-01-04

**Comment:**

We thank all reviewers for their time and thoughtful feedback. We have addressed the comments in a revised manuscript, and plan to submit to a future venue.

**Withdrawal Confirmation:**

I have read and agree with the venue's withdrawal policy on behalf of myself and my co-authors.